# Modeling the metabolic interplay between a parasitic worm and its bacterial endosymbiont allows the identification of novel drug targets

David M Curran[1†], Alexandra Grote[2†], Nirvana Nursimulu[1,3], Adam Geber[2], Dennis Voronin[4], Drew R Jones[5], Elodie Ghedin[2,6]*, John Parkinson[1,3,7,8]*

[1]Program in Molecular Medicine, Hospital for Sick Children, Toronto, Canada; [2]Department of Biology, Center for Genomics and Systems Biology, New York University, New York, United States; [3]Department of Computer Science, University of Toronto, Toronto, Canada; [4]New York Blood Center, New York, United States; [5]Department of Biochemistry and Molecular Pharmacology, New York University School of Medicine, New York, United States; [6]Department of Epidemiology, School of Global Public Health, New York University, New York, United States; [7]Department of Biochemistry, University of Toronto, Toronto, Canada; [8]Department of Molecular Genetics, University of Toronto, Toronto, Canada

**\*For correspondence:**
elodie.ghedin@nyu.edu (EG);
jparkin@sickkids.ca (JP)

[†]These authors contributed equally to this work

**Competing interests:** The authors declare that no competing interests exist.

**Abstract** The filarial nematode *Brugia malayi* represents a leading cause of disability in the developing world, causing lymphatic filariasis in nearly 40 million people. Currently available drugs are not well-suited to mass drug administration efforts, so new treatments are urgently required. One potential vulnerability is the endosymbiotic bacteria *Wolbachia*—present in many filariae—which is vital to the worm. Genome scale metabolic networks have been used to study prokaryotes and protists and have proven valuable in identifying therapeutic targets, but have only been applied to multicellular eukaryotic organisms more recently. Here, we present *i*DC625, the first compartmentalized metabolic model of a parasitic worm. We used this model to show how metabolic pathway usage allows the worm to adapt to different environments, and predict a set of 102 reactions essential to the survival of *B. malayi*. We validated three of those reactions with drug tests and demonstrated novel antifilarial properties for all three compounds.

## Introduction

Filarial nematodes are responsible for neglected tropical parasitic diseases that are among the leading causes of morbidity worldwide. One of the most debilitating is lymphatic filariasis (LF)—also called elephantiasis—which is caused by *Brugia malayi*, *Brugia timori*, and *Wuchereria bancrofti*. As of 2015, an estimated 38.8 million people had lymphatic filariasis with an estimated 1 billion people at risk in 72 endemic countries (*Vos and GBD 2015 Disease and Injury Incidence and Prevalence Collaborators, 2016*). Transmission occurs when microfilariae released by a female worm within an infected individual circulate in the blood where they are ingested by one of several species of mosquito. In the insect vector, the larvae undergo development; during subsequent blood meals, infective third-stage filarial larvae (L3) are transmitted to a mammalian host by penetrating the bite wound. They subsequently develop through molts into adult worms in the lymphatics (*Gleave et al., 2016*).

Current mass drug administration efforts involve a small selection of drugs—diethylcarbamazine, ivermectin, and albendazole—with limited effectiveness against the adult stages of the parasites. To prevent transmission and to relieve symptoms, treatment must continue for the lifespan of the adult worms, which can be up to 15 years (*Molyneux et al., 2014*). Furthermore, diethylcarbamazine is contraindicated in regions where *Onchocerca volvulus* or *Loa loa*—other filarial nematodes—are endemic (*Gyapong et al., 2005*; *Taylor et al., 2010*). Ivermectin is also contraindicated in regions where *L. loa* is co-endemic due to potential life-threatening complications (*Gardon et al., 1997*; *Boussinesq et al., 1998*; *Boussinesq et al., 2006*). While anthelmintic resistance has not yet manifested as a serious treatment issue in humans as it has in veterinary medicine (*Kaplan and Vidyashankar, 2012*), the potential remains a serious threat; emergence of resistance in these species against diethylcarbamazine (*Eberhard et al., 1991*), ivermectin (*Awadzi et al., 2004*; *Eng et al., 2006*), and albendazole (*Schwab et al., 2005*) has been reported for many years.

An alternative strategy for treatment has been the use of traditional antibiotics to target the endosymbiotic bacteria that live within most filarial nematodes. These bacteria are from the genus *Wolbachia*, specific to each helminth, and found to be essential for adult worm fitness and reproduction (*Taylor et al., 2013b*). Targeting these bacteria with the antibiotic doxycycline was shown to reduce numbers of *Wolbachia* present in the worms, sterilize adult females, and reduce symptoms of lymphatic filariasis (*Debrah et al., 2009*; *Rao et al., 2012*; *Ghedin et al., 2009*; *Taylor et al., 2010*). While antibiotic treatment remains a viable option for individual patients, long treatment times and contraindications for children and pregnant women limit its suitability for mass-drug administration efforts (*Taylor et al., 2010*). This strategy is ongoing, as recent evidence suggests that the commonly used antibiotic rifampicin may also possess filaricidal activity, but these data are preliminary and have not yet been tested in humans (*Aljayyoussi et al., 2017*). A more recent study proposes faster-acting antibiotics that belong to the tetracycline class of drugs (*Taylor et al., 2019*).

Considering the limitations of current treatment regimens, there is an urgent need to identify new drug targets for *B. malayi* that directly impact adult worm survival and, if possible, are specific enough to avoid the potential complications that arise in regions co-endemic for *O. volvulus* or *L. loa*. The bioinformatic identification of essential pathways as drug targets has been successfully demonstrated against many pathogens, including nematodes (*Taylor et al., 2013a*), *Entamoeba histolytica* (*Klebanov and Yakovlev, 2007*), *Plasmodium falciparum* (*Yeh et al., 2004*; *Fatumo et al., 2009*; *Chiappino-Pepe et al., 2017*), and various bacterial pathogens (*Rahman and Schomburg, 2006*; *Kim et al., 2010*; *Kim et al., 2011*). Genome scale metabolic reconstruction and constraint-based modeling in particular have emerged as effective strategies to identify critical metabolic enzymes and pathways (*Oberhardt et al., 2008*; *Chavali et al., 2008*; *Lee et al., 2009*; *Song et al., 2013*), which, due to their importance in energy production and in generating the building blocks required for growth and survival, are good potential therapeutic targets (*Cotton et al., 2016*; *Chiappino-Pepe et al., 2017*).

These models can be analyzed using flux balance analysis (FBA), an optimization method that is applied to a metabolic network (reviewed in *Orth et al., 2010*). Briefly, FBA calculates the maximum amount of biomass that can be produced given the available nutrients and the reaction constraints in the model, as well as the flux through each reaction needed to attain that solution. Beyond the identification of essential genes and potential therapeutic targets, the analyses of metabolic reconstructions with FBA have been used to identify knowledge gaps and improve annotations in pathogens like *Pseudomonas aeruginosa* (*Oberhardt et al., 2008*) and *Leishmania major* (*Chavali et al., 2008*), improve bioreactor yields of non-vital compounds in *Pseudomonas putida* (*Puchałka et al., 2008*), explain the complex observed substrate specificities of *Desulfovibrio vulgaris* (*Flowers et al., 2018*), explain observed metabolic changes in the brains of patients with Parkinson's disease (*Supandi and van Beek, 2018*), and even demonstrate the non-biomass related factors affecting tissues growing by cell expansion in tomato plants (*Shameer et al., 2020*).

Here, we describe the first metabolic reconstruction and constraint-based models of *B. malayi*. Using the high quality genome sequence of *B. malayi* (*Ghedin et al., 2007*; *Foster et al., 2020*), we first generated a network representation of the parasite's metabolic capabilities. We integrated previously published stage-specific transcriptome datasets for both *B. malayi* and its *Wolbachia* endosymbiont (*Grote et al., 2017*). We revealed stage-specific metabolic dependencies and identified enzymes that are predicted to be effective targets for drug intervention strategies. In subsequent drug inhibition studies, we validated three of these targets and show the novel antifilarial properties

of three human drugs. To our knowledge, this work represents the first compartmentalized metabolic model for any parasitic nematode.

## Results

### *i*DC625: The first genome scale metabolic model for *B. malayi*

To develop a genome scale metabolic model of *B. malayi*, we first developed a draft network as previously described (*Song et al., 2013*; *Blazejewski et al., 2015*; *Cotton et al., 2016*) (see Materials and methods for details). The draft network was then manually curated and divided into three compartments: the cytosol and mitochondria of *B. malayi*, and the *Wolbachia* endosymbiont.

In order to model *B. malayi* growth with pFBA (parsimonious FBA), we assembled a set of required biomass metabolites together with their relative abundances. This collection represents the biomass objective function and is used in constraints-based modeling to calculate flux distributions for each reaction in the model. Here we based the objective function on a previously defined function generated for *O. volvulus* (*Cotton et al., 2016*), modified with *B. malayi* and *Wolbachia*-specific values for DNA, RNA and amino acid distributions that were obtained from previously published studies (*Ghedin et al., 2007*; *Foster et al., 2005*; *Grote et al., 2017*) (see Materials and methods for the full function).

The final reconstruction, designated *i*DC625, contains 1266 total reactions involving 1252 total metabolites. Of the 1266 reactions, 1011 represent enzymes of which 849 are associated with 625 genes; 575 of these enzymatic reactions are associated with the cytosolic compartment, 166 with the mitochondria, and 270 with *Wolbachia* (*Figure 1*). Of the remaining reactions, 226 are associated with transport across compartments, of which 37 represent metabolite exchange between the mitochondria and cytosol, 80 between *Wolbachia* and the cytosol, and the remaining 109 representing metabolite exchange between the cytosol and the extracellular milieu. A further 29 reactions are artificial, used only to organize the biomass components required for the objective function (see Materials and methods for details). After filtering compartmental duplicates, the 1011 enzymatic reactions represent 761 unique KEGG reactions. For context, the *C. elegans* metabolic reconstruction *i*CEL1273 (*Yilmaz and Walhout, 2016*) contains 1985 total reactions representing 929 unique KEGG reactions; a comparison between these sets indicates that 319 (42%) of the KEGG reactions are unique to *B. malayi* and 442 (58%) are shared with *i*CEL1273. Of the 1252 metabolites, 661 are associated with the cytosol, 202 with the mitochondrion, 362 with the *Wolbachia*, and 27 are used in the artificial conversions of biomass components. These represent 1025 unique KEGG metabolites, of which 602 (59%) are unique to *B. malayi* and 423 (41%) are shared with *i*CEL1273.

We also used life stage-specific gene expression data to constrain the reactions of *i*DC625. In total, we obtained relative expression data for 11,840 *B. malayi* and 823 *Wolbachia* genes across ten different life-stages; 87.6% of the *B. malayi* genes and 96.4% of the *Wolbachia* genes were expressed in at least one stage. This yielded 11 total models: unconstrained (open), L3, L3 6 days post-infection (L3D6), L3 9 days post-infection (L3D9), L4, adult female 30 days post-infection (F30), adult female 42 days post-infection (F42), adult female 120 days post-infection (F120), adult male 30 days post-infection (M30), adult male 42 days post-infection (M42), and adult male 120 days post-infection (M120).

### *Wolbachia* weight impacts model performance

Since the presence of *Wolbachia* directly impacts model dynamics, both through the production of metabolites that contribute to the biomass objective function as well as the consumption of metabolites to maintain its own growth, the relative weight between bacteria and worm must be considered in the model. This weight is implemented by weighting the *Wolbachia* contribution to the biomass objective function (see Materials and methods), as well as constraining reactions assigned to the endosymbiont. Using these constraints in a pFBA framework, we examined how changes in *Wolbachia* weights affect the maximum flux through the objective function of the model. As the availability of a carbon source and of oxygen are two of the most important determinants of the model's activities, we examined the model under four different nutrient conditions: high oxygen (580 units) and high glucose (250 units) (HOHG), high oxygen and low glucose (45 units) (HOLG), low oxygen (90 units) and high glucose (LOHG), and low oxygen and low glucose (LOLG). These units are not



**Figure 1.** The *iDC625* genome scale metabolic model. Network representation of our metabolic reconstruction, where the large colored circles represent reactions that are connected by metabolites in light grey.

calibrated to any real-world quantities, so our choices were inherently arbitrary. The low values were as low as possible where metabolic pathways were functional and relatively stable; these pathways were utilized in different ways as levels of both nutrients were increased, so our high values were chosen after the point where pathway utilization became stable again.

By applying these constraints to the otherwise unconstrained model and varying the relative weight of *Wolbachia*, we found that the maximum production of the biomass objective function occurs at *Wolbachia* weights of 0.04, 0.02, 0.02, and 0.01 under HOHG, HOLG, LOHG, and LOLG, respectively (*Figure 2*).

The behavior of the life stage models under changing weights was relatively similar, except for F120 and M120, which had by far the least biomass production. In all cases the models were more sensitive to increasing the *Wolbachia* weight under low oxygen than under low glucose conditions, as evidenced by the relative gradients associated with LOHG compared to HOLG (*Figure 2*). Interestingly, the maximum objective function flux under both HOHG and HOLG conditions were

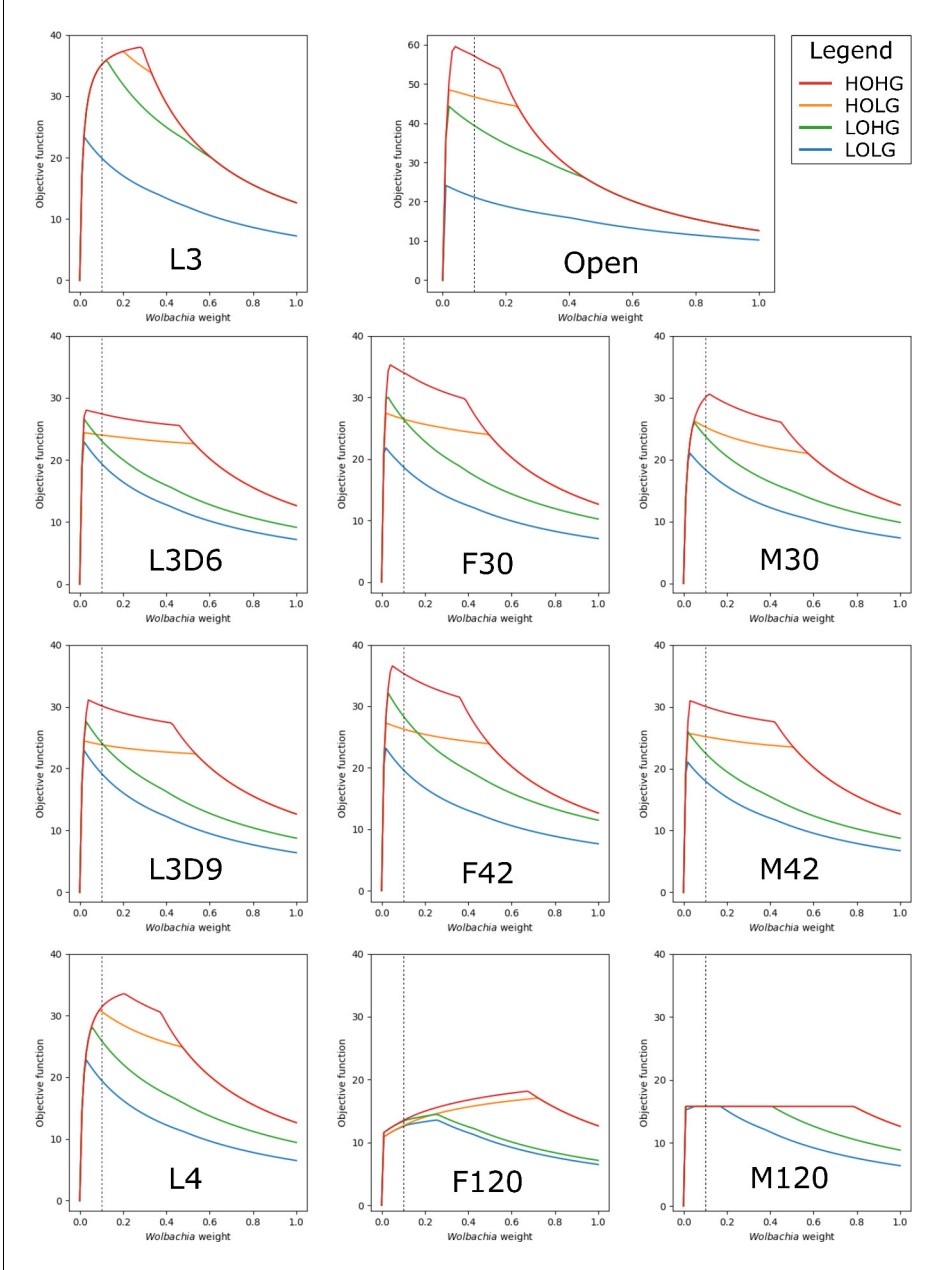

**Figure 2.** The effect of the *Wolbachia* weight on the maximum objective function flux under different nutrient conditions and life stages. The life stage models were generated by mapping different stage-specific expression data on the unconstrained (Open) model: larval stage 3 (L3), L3 6 days post-infection (L3D6), L3 9 days post-infection (L3D9), L4, adult female 30 days post-infection (F30), adult female 42 days post-infection (F42), adult female 120 days post-infection (F120), adult male 30 days post-infection (M30), adult male 42 days post-infection (M42), and adult male 120 days post-infection (M120). The dashed line indicates the *Wolbachia* weight of 0.1, which is used in all subsequent experiments.

The online version of this article includes the following figure supplement(s) for figure 2:

**Figure supplement 1.** The effect of the *Wolbachia* weight on the maximum objective function flux when supplemented with excess pyruvate.

identical for all eleven models at the maximum *Wolbachia* weight tested (***Supplementary file 1*** - Supplemental Table 1); this indicates that the effects of a very high *Wolbachia* weight overwhelm the life stage-specific reaction constraints as well as any benefits from an excess carbon source. Under all conditions the model was unfeasible with no *Wolbachia* (which agrees with the experimentally determined essential role of the endosymbiont), and as the *Wolbachia* weight increased the objective function peaked rapidly followed by a steady decline. Since the negative impact to the objective function was minimal until 0.18, we assigned a weight of 0.1 under all nutrient conditions. This was chosen to minimize the impact on the objective function, while still being conservative in allowing reasonable flux through *Wolbachia* reactions.

As *Wolbachia* population dynamics have been well-studied in *B. malayi* (***McGarry et al., 2004***; ***Grote et al., 2017***), it is tempting to attempt to calibrate our model to life stage-specific population sizes. However, our model is concerned with the total metabolic capacity of the bacterium, which may not correlate directly with population size, and there is very little known about the nutrients available in the environments occupied by many of the life stages, especially metabolically inert stages like L3 (***Li et al., 2009***). Further, we cannot directly measure the change in *Wolbachia* in response to changing conditions because our models are designed such that the *Wolbachia* weight is a parameter in the biomass optimization, not an outcome of that function.

One approximation is to compare the model performance at a high *Wolbachia* weight between different conditions, which indicates how well a model is able to tolerate the *Wolbachia*-associated resource drain, though the conditional structure of this measurement is reversed compared to a direct measurement. With this in mind, we attempted to validate our models against a previous finding that *Wolbachia* populations expanded in adult male *B. pahangi* worms when supplemented with exogenous pyruvate (***Voronin et al., 2019***; ***Figure 2—figure supplement 1***). At a *Wolbachia* weight of 1.0, the objective function averaged over HOHG, HOLG, LOHG, and LOLG increased for all life stage models when excess pyruvate was made available (***Supplementary file 1***-supplemental table 1: 'The objective flux under a high *Wolbachia* weight'). The increase was largest in the three adult male models (10.3%), compared to adult females (6.16%), the larval stages (7.20%), or the unconstrained model (5.88%). This effect was not simply an artifact of our choice of *Wolbachia* weight, as the same trend was observed when measuring the maximum flux achieved under any *Wolbachia* weight with and without the addition of excess pyruvate (***Supplementary file 1***-supplemental table 2: 'The maximum objective flux achieved with varying *Wolbachia* weight').

## Altering nutrient conditions reveals a metabolic landscape rich in alternative energy production pathways

Over the course of its lifecycle, *B. malayi* encounters a range of different nutrient conditions and likely regulates enzyme expression to alter metabolic flux to optimize growth in each condition. For example, adult worms are found in the lymphatic system where the expectation is that the parasite is exposed to a substantially lower oxygen environment compared to earlier life stages like the L3. We therefore performed a series of pFBA simulations in which we examined the impact of changes in two key metabolites on worm growth, oxygen and glucose, in the different life stage models (***Figure 3***). Fumarase is one of the measured reactions and is part of the tricarboxylic acid (TCA) cycle that converts fumarate into malate. In addition to being located directly downstream of Complex II in the TCA cycle, it is also directly upstream of the anaerobic reverse Complex II (***Figure 4***). Given that fumarase is reversible, flux measurements provide an indication of the activity of both aerobic and anaerobic metabolism. As expected, pFBA predicts a reliance on anaerobic pathways under low oxygen conditions (***Figure 3***). However, at an oxygen concentration of 205 flux units (***Figure 3***, vertical-dashed line), there is a switch to the aerobic pathway in the unconstrained and L3 models. Interestingly, the acetate and propanoate waste transporters are no longer used above this threshold; the predicted presence of these waste products is consistent with previous studies of helminth anaerobic metabolism (***Tielens et al., 2010***; ***Müller et al., 2012***).

To further explore the impact of glucose and oxygen availability on *B. malayi* metabolism, we conducted detailed analyses under the four diverse nutrient conditions previously described: HOHG, HOLG, LOHG, and LOLG. For each condition, the model was provided with sufficient fatty acids (50 units), amino acids (100 units), and several cofactors: ammonia, phosphate, H⁺, adenine, pyridoxal phosphate, heme, folate, cholesterol, oleic acid, pantothenate, choline, riboflavin, putrescine,



**Figure 3.** Reaction fluxes when varying oxygen and glucose. The activities of the objective function and five other reactions are shown for a range of 25 oxygen and 25 glucose availabilities for each life stage model. The life stage data were generated by mapping different stage-specific expression data on the unconstrained (Open) model: larval stage 3 (L3), L3 6 days post-infection (L3D6), L3 9 days post-infection (L3D9), L4, adult female 30 days post-infection (F30), adult female 42 days post-infection (F42), adult female 120 days post-infection (F120), adult male 30 days post-infection (M30), adult male 42 days post-infection (M42), and adult male 120 days post-infection (M120). For each, pFBA was performed for all 625 combinations of oxygen and glucose; the color of each pixel in the heatmap indicates the activity of the reaction at that nutrient availability, with white, dark blue, and dark red indicating no activity, maximum forward, and maximum reverse, respectively. As each reaction has a different activity profile, each has its own color legend. The four black boxes on each graph indicate what we discuss as low and high concentrations of oxygen (90 and 580 units) and glucose (45 and 250 units), and the vertical dashed line indicates the anaerobic-aerobic threshold at 205 units of oxygen.



**Figure 4.** Schematic diagram of the major catabolic pathways observed in the model. Metabolites were manually tracked in the model under many different nutrient conditions and life stage expression constraints, and the major catabolic pathways are diagrammed here. The thick dark line indicates the mitochondrial/*Wolbachia* membrane, and those pathways present in the *Wolbachia* compartment are indicated by (W). The labeled metabolites are Glc: glucose; Pep: phosphoenolpyruvate; Pyr: pyruvate; Acc: acetyl-coenzyme A; Fat: fatty acids; Cit: citrate; Ict: isocitrate; Akg: alpha-ketoglutarate; Scc: succinyl-coenzyme A; Suc: succinate; Fum: fumarate; Mal: malate; Oaa: oxaloacetate; Glu: glutamate; Asp: aspartate; Ace: acetate; Prp: propanoate. The labeled reactions are CpxI: Complex 1; CpxII: Complex 2; CpxIII: Complex 3; CpxIV: Complex 4; CpxV: Complex 5; Glyox: glyoxylate. The energy transfer molecules are common metabolites, where a blue border indicates that the molecule was produced, and a red border indicates that it was consumed. The labeled energy transfer molecules are A: ATP; G: GTP; N: NADH; U: ubiquinol; R: rhodoquinol; H: proton-motive force; O: oxygen.

nicotinate, UTP, CTP, $Fe2^+$, and N-acetyl-D-glucosamine; in all simulations, the maximum usage of fatty acids, amino acids, and cofactors was less than provided.

We observe that the model produces the most biomass under HOHG conditions, as expected. Instead of using the full TCA cycle, the model predicts a reliance on the glyoxylate shunt to produce malate and succinate (*Figure 4*). A large proportion (74.6%) of malate is processed by the TCA cycle to isocitrate which is recycled back into the glyoxylate shunt, and 18.2% is predicted to be

processed through anaerobic pathways to produce succinate, a phenomenon termed the Crabtree effect (*Postma et al., 1989*; *Pfeiffer and Morley, 2014*), in which anaerobic pathways are used by the cell in the presence of high oxygen and glucose. Succinate produced through both pathways is subsequently exported from the cell as waste. The majority of the oxygen imported (74.5%) is used to generate ATP through oxidative phosphorylation, while only 6.2% (36 units) is predicted to be used by *Wolbachia*.

Under HOLG conditions, growth is predicted to fall to 82% of optimal biomass production. While the glyoxylate shunt is still used, succinate is no longer exported, but instead is processed by the TCA cycle to produce energy, with $CO_2$ subsequently exported as waste. Similar to HOHG, 71.2% of the oxygen is used for oxidative phosphorylation in the mitochondria, while 6.0% (35 units) is used by *Wolbachia*.

Under LOHG conditions, biomass production falls to 69%. Here, the glycolytic pathway is used to generate energy, with glucose metabolized to phosphoenolpyruvate (PEP) producing approximately half of the total NADH used by the model, which is then transported into the mitochondria via the malate-aspartate shuttle. The other half of the NADH is produced by the classical anaerobic pathway involving the conversion of some PEP (37.3%) to pyruvate, which is subsequently transported into the mitochondria, metabolized to acetate, and excreted as waste. However, most of the PEP (59.0%) is processed by the nematode-specific anaerobic pathway. Through this pathway, PEP is metabolized into malate and transported to the mitochondria where it is processed to succinate by the reverse Complex II, and ultimately converted to propanoate and exported as waste. Under these conditions, most of the oxygen in the model (61.0%) is used to generate ATP through oxidative phosphorylation, while 37.8% (34 units) is used by *Wolbachia*.

Under LOLG conditions biomass production is reduced to 37% of the optimal. Glucose is converted to malate and transported into the mitochondria as observed under LOHG conditions. Approximately half is processed to succinate by the reverse Complex II, while the other half is used to generate NADH via conversion to oxaloacetate, α-ketoglutarate (AKG), succinyl-CoA, and then succinate. All of the succinate is then converted to propanoate and exported. Under these conditions 69.8% of the oxygen is being consumed by oxidative phosphorylation, and 29.6% (27 units) is used by *Wolbachia*.

Interestingly, the model predicts a potentially novel form of glutamate metabolism, representing a combination of the malate-aspartate shuttle and the TCA cycle (*Figure 4*). It is similar to the utilization of malate under LOLG conditions, except that succinate is recycled by the TCA cycle instead of being exported. In this pathway, glutamate is first transported into the mitochondria in exchange for aspartate, before combining with oxaloacetate to yield AKG and aspartate. AKG is then processed by the TCA cycle to regenerate oxaloacetate. The net reaction results in the conversion of glutamate to aspartate and $CO_2$, along with the production of key energy metabolites: GTP, 2 x NADH, and ubiquinone. This is almost equivalent to the energy produced by the catabolism of a single molecule of acetyl CoA by the TCA cycle: 2 x $CO_2$, GTP, 3 x NADH, and ubiquinone. Given that the glutamate/aspartate transporter is driven by the proton-motive force, it is not known if this pathway would be energetically favourable *in vivo* (*Bremer and Davis, 1975*; *Bakker et al., 2001*). However, the striking resemblance to the catabolism used by intestinal epithelial cells (*Blachier et al., 2009*) and to the relatively recently described 'glutamine addiction' pathway used by many cancer cells (*Wise and Thompson, 2010*; *Mazat and Ransac, 2019*) suggests it may be physiologically relevant.

It has previously been suggested that *Wolbachia* may function to supplement mitochondrial energy production in filarial nematodes (*Darby et al., 2012*); our model predictions support this hypothesis. Under LOLG conditions, *Wolbachia* is predicted to export the maximum amount of ATP possible (100 units) into the *B. malayi* cytosol. It was also suggested that *Wolbachia* uses pyruvate as its primary carbon source (*Voronin et al., 2016*), but under LOLG conditions pyruvate import is nearly zero, and *Wolbachia* is using the novel glutamate metabolic pathway described above. Under LOHG conditions, *Wolbachia* ATP export drops to 90% compared to LOLG conditions. Pyruvate import increases substantially (nine units instead of 1), and the model uses both the TCA cycle and the glutamate metabolic pathway to generate energy. The *Wolbachia* metabolic pathways appear much the same under the other conditions, except that ATP export drops to 71%, and 44% of LOLG for HOLG and HOHG, respectively.

## Life stage specific metabolic models of *B. malayi* reveal a dynamic reliance on alternative pathways

To determine how metabolic pathway dependencies may vary across the *B. malayi* life cycle, pFBA was performed under the four different nutrient conditions for each of the ten life cycle stage models (*Figure 5*). As expected, the models produced the most biomass under HOHG conditions, and the unconstrained (open) model produced the most under all conditions. In general, models are able to produce more biomass when presented with additional nutrients, which is why biomass production increases when moving from low glucose to high glucose, or from low oxygen to high oxygen conditions.

There was little difference under LOLG conditions between any of the models, except for adult worms at 120 dpi (M120 and F120). This indicates that the different reaction constraints are playing a minor role under these conditions, and that the concentrations of glucose and oxygen are the limiting factors.

While the L3 model experiences a large benefit under LOHG or HOLG conditions compared to LOLG, there is no increased benefit when moving from either to HOHG conditions. This indicates that the reaction constraints imposed for this stage limit the model's ability to exploit increases in both nutrients, in contrast to the open model which sees biomass increasing when moving from either LOHG or HOLG to HOHG. The M120 model appears to be saturated for these nutrients even under LOLG conditions, as its biomass production never increases when more of either nutrient is made available. The F120 model appears to be nearly saturated and receives only a modest benefit from additional glucose or oxygen.

Besides measuring the biomass production for each model under different nutrient conditions, we also quantified the number of reactions used by the models (*Figure 6*). We found that the *Wolbachia* compartment changes the least among the different models and conditions, while the mitochondrial compartment shows the most variation. This also allows us to identify 'enzymatically constrained' models, which is a term that relates to the number of possible ways a model is able to achieve its maximum objective flux; if there are multiple alternate metabolic pathways that can be used to satisfy the objective function, a model would be considered enzymatically unconstrained. Models generally become less constrained as they are provided with more nutrients.

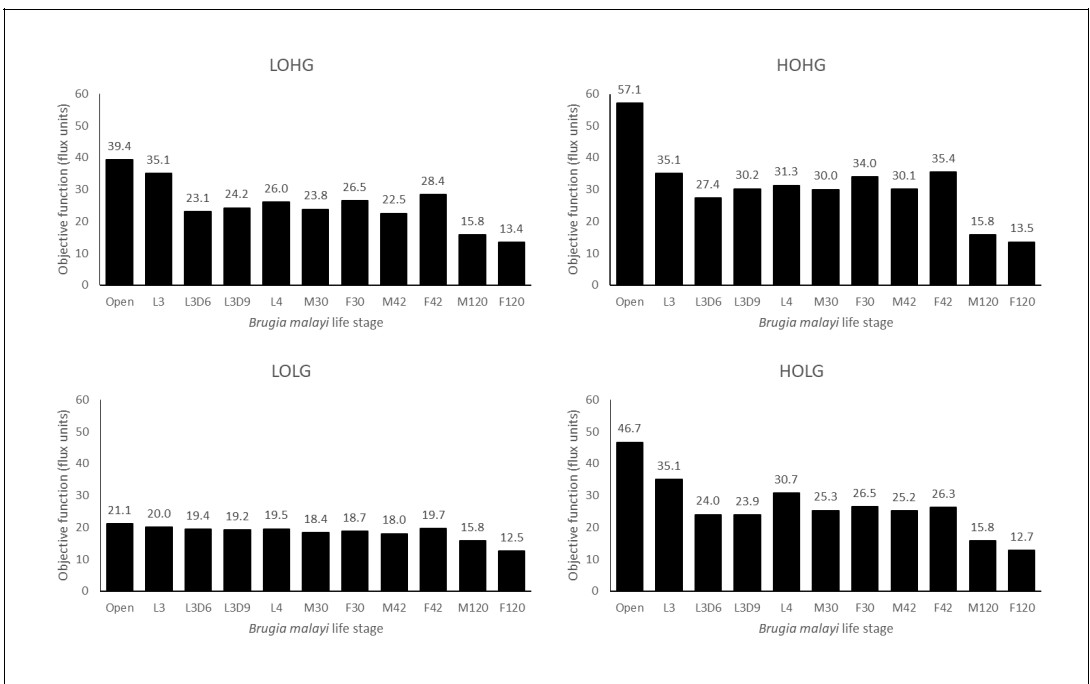

**Figure 5.** Biomass production across life stages under different nutrient conditions. The maximum biomass produced by each life stage model under LOLG, LOHG, HOLG, and HOHG conditions.



**Figure 6.** Reaction utilization across life stages. The number of reactions used by each model in each compartment to achieve its maximum objective function flux, under A) LOHG, B) HOHG, C) LOLG, and D) HOLG conditions. The black bars indicate the number of reactions used in the most parsimonious solution, while the stacked white bars indicate the number of reactions used in all possible solutions that yield the same value for the objective function. The larger the height discrepancy between the two bars, the more redundant pathways the model has available to achieve the same objective function flux.

It is interesting to note that both the L3D6 and L3D9 models appear to be substantially more enzymatically constrained than L3. These two stages represent the time when *B. malayi* is molting, specifically during the apolysis stage of molting where a new cuticle is being synthesized to replace the old cuticle. These reduced usage numbers, combined with reduced biomass production (*Figure 5*), may reflect this specialized function and large energetic expenditure. Differences between these related models extend to pathway utilization as well. Under HOHG conditions, the L3 model uses the whole TCA cycle—though fatty acids are used as the sole carbon source instead of glucose—while the other two stages exhibit the apparent Crabtree effect. Unlike the unconstrained model, none of these three larval-stage models export any significant amount of succinate. Additional differences include L3D9 exporting a large quantity of acetate, and L3D6 and L3D9 exporting propanoate (*Figure 3*).

## Metabolomics data identify intermediates throughout, at the lowest levels in L3

To help validate our reconstruction, we performed untargeted metabolomics on worm extracts from four different life cycle stages. Applying two complementary approaches, our analyses identified 146 unique metabolites from a 'hybrid' analysis and 492 from a 'predictive' analysis (see Materials and methods for details); of these, 103 and 316 metabolites were found in at least three samples, respectively. Directly relating these data to the model predictions is unfortunately non-trivial, as the metabolomics analyses detect the size of the pool of some metabolite at one point in time, while our pFBA predictions are steady-state rates.

Most of the TCA cycle intermediates (*Figure 4*) were detected in one or both of our metabolomics data sets. Phosphoenolpyruvate was only detected in adults in the hybrid analysis but was found in the microfilaria and L3 stages as well as in the predicted analysis. Pyruvate was detected only in the adult worms in the hybrid analysis, and at the highest levels in males. Oxaloacetate was detected in all stages in the predicted analysis, but at the lowest levels in the adults. Citrate and isocitrate were detected in the adults and one microfilaria sample in the predicted analysis, and only in the adults in the hybrid analysis. Alpha-ketoglutarate was detected in both analyses in the adults and one microfilaria sample, highest in the females. Succinate, fumarate, and malate were detected in the adults and microfilaria, with none in the L3 samples. This pattern was consistent in both analyses, except for succinate which was missing from the hybrid analysis. No acetyl-CoA or succinyl-CoA were detected in either analysis.

We were able to detect several fatty acid degradation intermediates as well, in particular carnitine-conjugated fatty acids. Carnitine and acetylcarnitine were detected in all samples in both analyses and were found to be highest in males, followed by females, microfilaria, and finally L3. Other short-chain acylcarnitines, butyrylcarnitine and valerylcarnitine were detected in all samples except L3 in the hybrid analysis. Consistent with above, these metabolites were found to be highest in males, followed by females, and then microfilaria. Propionylcarnitine was detected only in the adult samples in the hybrid analysis. The only long-chain fatty acid intermediate detected was palmitoylcarnitine, and only in the microfilaria and one male sample in the predicted analysis.

If we accept that higher levels of these intermediates imply increased metabolic activity, then we observe a general trend of the highest activity in the adults (in particular males metabolizing fatty acids), followed by the microfilaria, with L3 having the lowest levels if any were detected. This fits with the observation that L3 is a non-feeding stage, which has been likened to the dauer stage of *C. elegans* (*Li et al., 2009*). Though we were not able to use these metabolomics data to quantitively validate our models, we include them here as they may prove useful to the wider research community (*Supplementary file 2*: Metabolomic analysis of four *Brugia malayi* life stages).

## Modeling of *B. malayi* metabolism predicts novel therapeutic targets

To identify critical reactions in the metabolic network that represent potential therapeutic targets, we performed a series of *in silico* knockouts in which each reaction constraint was set to zero (i.e. no flux was allowed through that reaction). Of the 1011 enzymatic reactions in *i*DC625, 815 had no impact on the biomass objective function when knocked out (*Figure 7A*), 94 were found to have a low impact (biomass ≥50% of baseline), and 102 were found to be essential (biomass <50% of baseline). The biomass breakpoints for no impact and essential were actually set to within 0.0001 flux units (<0.0002% of the biomass) of the true values to account for floating point errors.

This set of essential reactions was generated under HOHG conditions, but proved to be quite robust to the model conditions. Increasing the *Wolbachia* weight from 0.1 to 0.5 decreased the maximum biomass production by ~20% but had no effect on the reactions predicted as essential, and even increasing it to 0.9 only reclassified one reaction as non-essential (KEGG reaction R07618 from the *Wolbachia* compartment). The only change under LOLG conditions was to reclassify this same reaction as non-essential. Under HOLG conditions three new reactions were classified as essential (R00081, R00086, and R02161 from the mitochondrial compartment), though these same reactions from the *Wolbachia* compartment were already classified as essential. Four new essential reactions were identified under LOHG conditions (R00658, R01061, R01512, and R01518 from the cytosolic compartment), though again these same reactions from the *Wolbachia* compartment were already classified as essential.



**Figure 7.** Essential reactions in the model. (**A**) Breakdown of the essentiality of the model reactions by their compartment. No impact indicates the model's optimized biomass objective function was unaffected, low impact indicates biomass was ≥50% of baseline, and essential indicates that biomass was <50% of baseline. (**B**) A Venn diagram of the overlap in predicted essential reactions between our model, the previously published iCEL1273 *C. elegans* metabolic reconstruction, and experimentally determined essential reactions in *C. elegans*. All three sets of reactions represent only those that are present in both iDC625 and iCEL1273. (**C**) A heatmap showing the effects of double knockout combinations of 129 reactions that resulted in biomass <50% of baseline. A value of 1.0 (light yellow) indicates that there was no effect from the knockout, while a value of 0.0 (black) indicates that the model was unable to produce any biomass. Pathways are defined as containing one or more reactions, and 16 of the major pathways are labeled on the heatmap (P1 – P16). The legend on the right describes the interactions observed between the pathways, and their general functions.

Our set of essential reactions was compared to those predicted in the *iCEL1273 C. elegans* model, as well as experimentally determined essential reactions (*Yilmaz and Walhout, 2016*). The authors used the same definition for essential reactions, selecting those that reduced the model's biomass by 50% or more; of the 102 reactions we predict to be essential, 71 were found in the *iCEL1273* model. They predicted 159 essential genes, corresponding to 125 KEGG reactions, 56 of which are also found in *iDC625*. They also list 461 *C. elegans* genes that have been experimentally determined to be essential; these correspond to 463 KEGG reactions, 273 of which are also found in *iDC625*. There were only 14 essential reactions shared by *iCEL1273* and *iDC625* (*Figure 7B*), but this low overlap is likely because both sets of model predictions are missing many biologically essential reactions. This is suggested by the large significant overlap between each model's predictions and the experimentally determined essential reactions; even though they are separated by 400 million years of evolution and possess different life cycle strategies (parasitic v free-living), 73% of the predicted *iDC625* essential reactions overlap with the experimentally determined essential reactions of *C. elegans* (significance determined by one-tailed Fisher's Exact Test; hypergeometric p-value = $1.9E^{-11}$).

In addition to single reactions, we also investigated reaction pairs exhibiting synthetic lethal relationships. Such relationships may exist, for example, when two reactions operate in alternative pathways that can each lead to production of the same key metabolite. This 'two hit' strategy may offer greater long-term potential through the development of combination therapies that ultimately reduce the risk of emergence of resistance, both through requiring the pathogen to simultaneously acquire resistance to two independent targets, and through the use of lower drug dosages that can result from increased efficacy (*Lehár et al., 2009*; *Ejim et al., 2011*; *Spitzer et al., 2011*; *Aziz et al., 2015*).

Of the 909 reactions which were predicted to inhibit growth by less than 50% when knocked out individually, 129 were involved in at least one pair of knockouts that together reduced biomass production to less than 50% (*Figure 7C*). Analyses reveal that the model possesses alternative pathways to produce nicotinate—important in redox reactions—and that knocking out different combinations of these pathways had a dramatic impact on biomass production (*Figure 7C*; P1, P2, P3, and P11). Our simulations also predict that the loss of one half of the TCA cycle or the other can be compensated for by the model, but not both, and only if the *Wolbachia* TCA cycle is functional (*Figure 7C*; P4, P5, P6, and P15). While purine biosynthesis pathways are predicted to be essential by our *in silico* single knockouts, pyrimidine biosynthesis pathways are only predicted as essential through synthetic lethal interactions, suggesting redundancy in these pathways (*Figure 7C*; P7, P8, P9, and P10). Finally, we saw evidence of some redundancy in the *B. malayi* pentose phosphate pathways (*Figure 7C*; P12 and P13), and observed an interesting, nearly-lethal interaction between part of the pentose phosphate pathway involved in the metabolism of fructose 6-phosphate and the mitochondrial oxidative phosphorylation reactions (*Figure 7C*; P14 and P16).

## Fosmidomycin, MDL-29951, and Tenofovir possess antifilarial activity

To validate the performance of our model, we selected a subset of reactions for targeted inhibition using known drugs. Of the 102 reactions predicted to be essential (<50% baseline biomass production), 80 were associated with one or more genes (see *Supplementary file 3*: List of essential *Brugia malayi* genes, and prioritization information for details), of which 77 resulted in no biomass production when knocked out *in silico* (33 in the cytosol, 41 in *Wolbachia*, and three in the mitochondria). This subset was chosen because they were considered less likely to be model artifacts. Reactions were first ranked by the number of inhibitors to their cognate protein identified in the ChEBML database (*Supplementary file 3*: List of essential *Brugia malayi* genes, and prioritization information;

**Table 1.** Details about the three drugs tested for anti-filarial activity against *B. malayi* adult worms.

| Drug | Predicted target pathway | Developed for | Concentration (µM) |
| --- | --- | --- | --- |
| Fosmidomycin | Isoprenoid precursor biosynthesis | Antibiotic/antimalarial | 12.5 |
| MDL-29951 | Gluconeogenesis | Epilepsy | 12.5 |
| Tenofovir | Purine metabolism | Hepatitis B | 12.5 |

**Table 2.** Expression of the predicted drug target genes across *B. malayi* life stages.

|  | Target gene | Life stage expression (FPKM) | | | | | | | | | |
|---|---|---|---|---|---|---|---|---|---|---|---|
|  |  | L3 | L3D6 | L3D9 | L4 | F30 | F42 | F120 | M30 | M42 | M120 |
| Fosmidomycin | Wbm0179 | 12 | 17 | 18 | 11 | 27 | 11 | 16 | 37 | 31 | 3 |
| MDL-29951 | Bm13850 | 45 | 21 | 22 | 30 | 22 | 24 | 6 | 31 | 17 | 14 |
|  | Wbm0158 | 68 | 66 | 69 | 49 | 27 | 40 | 80 | 29 | 40 | 95 |
| Tenofovir | Bm9070 | 0 | 0 | 0 | 0 | 0 | 1 | 1 | 1 | 4 | 26 |
|  | Wbm0321 | 34 | 13 | 14 | 4 | 5 | 3 | 15 | 0 | 18 | 6 |
| Tenofovir-associated | Bm3965 | 18 | 19 | 20 | 11 | 57 | 33 | 14 | 15 | 19 | 17 |
|  | Bm14014 | 419 | 101 | 119 | 105 | 76 | 93 | 21 | 75 | 49 | 32 |

*Gaulton et al., 2012*; *Davies et al., 2015*; *Gaulton et al., 2017*). Hits lacking gene expression data were filtered out, as were those with more than three putative human homologs. We then prioritized hits based on the number and quality of published studies on those inhibitors, with extra consideration given to human clinical trials. This resulted in a short list of potentially druggable reactions: R01068, R00036, R05688, R00127, R00762, R05637/R05633, R04560/R01127, R00178, and R01920/R02869. We did not pursue R01068 as we have previously demonstrated significant antifilarial activity when its cognate gene is knocked down with RNAi (*Voronin et al., 2016*), nor did we pursue R00036 as it has been previously investigated by other groups as an antifilarial (*Lentz et al., 2013*). Due to the limited availability of adult *B. malayi* worms for testing, we assayed three of the predicted inhibitors (*Table 1*). They were chosen as they had all undergone previous clinical testing in humans and were available for order at a reasonable price from suppliers without requiring custom chemical synthesis. Future work may test additional inhibitors.

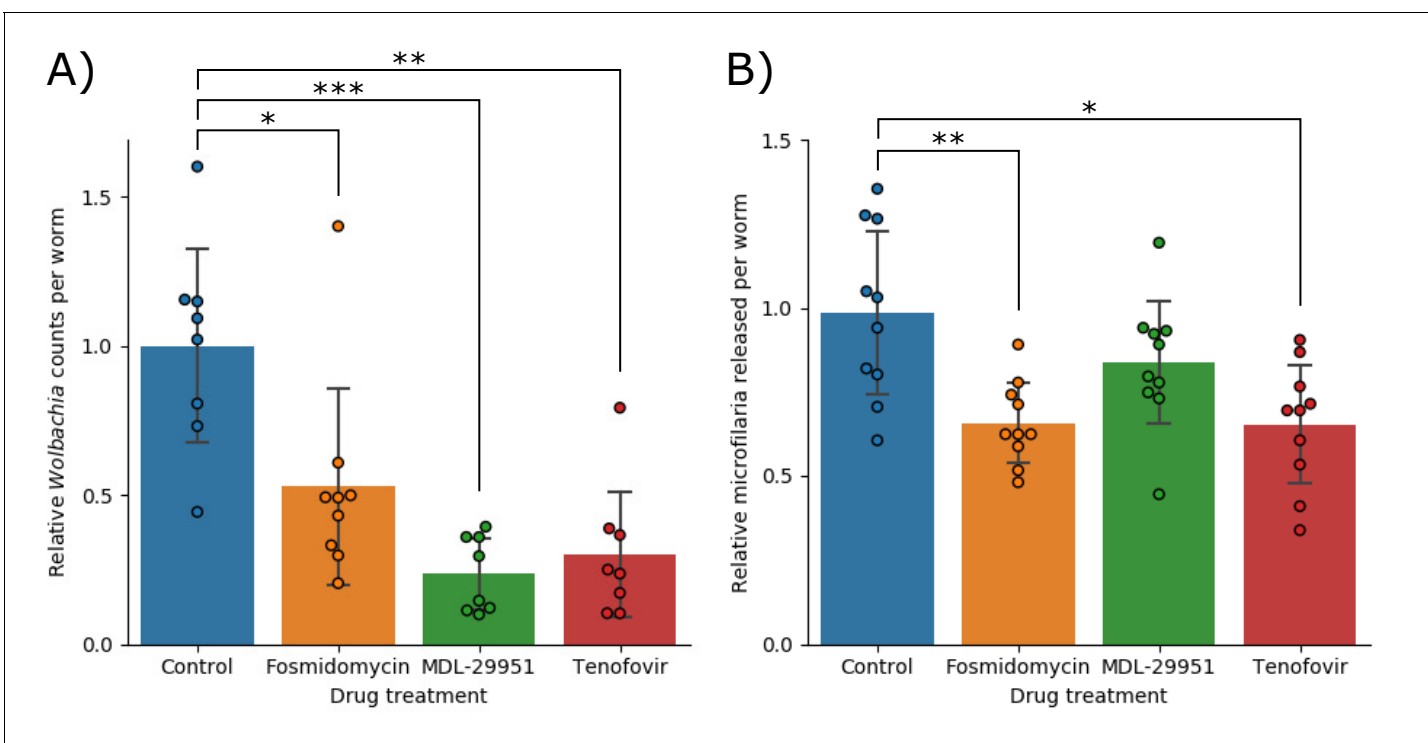

**Figure 8.** Anthelmintic activity against adult *B. malayi* worms. (**A**) Shows the number of *Wolbachia* detected per worm, normalized against the control group and (**B**) shows the number of microfilariae produced per worm, normalized against the control group. In both data sets significance was detected using a single factor ANOVA, followed by 2-tailed t-tests between each drug and the control with a Bonferroni correction. Error bars indicate the standard deviations; * indicates $p<0.05$; ** indicates $p<0.01$; *** indicates $p<0.001$.

As mentioned above, reactions were only prioritized as potential targets if their cognate genes were expressed across the *B. malayi* life stages (*Table 2*). The predicted Fosmidomycin target, *Wolbachia* gene Wbm0179 (1-deoxy-D-xylulose 5-phosphate reductoisomerase) has moderate expression that is consistent across most stages, high in early adult males but dropping off over time. The target gene of MDL-29951, Bm13850 (fructose biphosphatase), has moderate expression that is highest in L3 but relatively consistent across stages. *Wolbachia* also has a homolog of this gene, Wbm0158, that is expressed at a higher level especially in the late adult stages. Tenofovir was identified as a ChEMBL hit of *B. malayi* Bm9070 (an adenylate kinase), which has very low expression in most stages, but moderate in adult males. There is a *Wolbachia* homolog of this gene as well, Wbm0321, with low expression throughout the life stages. There are two other *B. malayi* genes that are associated with this reaction: Bm3965 (a UMP-CMP kinase) has moderate expression, highest in adult females but quite consistent across stages; and Bm14014 (adenylate kinase isoenzyme 1) has very high expression, highest in L3 but still high in male and female adults.

These three predicted drug targets were subsequently validated by testing their effects on worms *in vitro*. Fosmidomycin, MDL-29951, and Tenofovir were found to reduce the number of *Wolbachia* per worm to 53% ($\pm$35%; p=0.041), 24% ($\pm$13%; p=$7.2\times10^{-4}$), and 30% ($\pm$23%; p=0.0013) of control, respectively (*Figure 8*). We also observed two of the drugs impacting fecundity, with Fosmidomycin and Tenofovir reducing the number of microfilariae produced per worm to 66% ($\pm$12%; p=0.0091) and 65% ($\pm$18%; p=0.013) of control, respectively. Fosmidomycin treatment also appeared to lead to a consistent phenotype affecting motility, but this was not efficiently detected with the Worminator assay (*Supplementary file 1*-supplemental table 3: 'Motility of *B. malayi* was not affected by drug treatment').

## Discussion

We present the first constraints based metabolic model for *B. malayi*, which we term *i*DC625. While the model captures many known features of nematode metabolism, simulations under a variety of different conditions yielded a number of emergent behaviours, including switching between aerobic and anaerobic metabolic pathways, a predicted Crabtree effect under high oxygen and glucose, and a novel pathway that relies on the catabolism of glutamate to aspartate to generate energy. This suggests that in addition to being robust, the model is capable of generating novel biological hypotheses.

A high quality compartmentalized metabolic model allows us to study the metabolic processes of the cell in detail, including pathways that have been poorly studied in the past; in particular, the anaerobic metabolic pathways used by parasitic nematodes are unlike those studied in most other eukaryotes (*Del Borrello et al., 2019*). Our model is the first to incorporate this pathway and is therefore likely to yield accurate predictions as low oxygen environments are biologically relevant for parasitic nematodes.

An emergent behaviour predicted by the model was the exhibition of the Crabtree effect, a previously described phenomenon observed in yeast where anaerobic fermentation pathways are activated in aerobic conditions, but only in the presence of high levels of glucose (*Postma et al., 1989*; *Pfeiffer and Morley, 2014*). It is interesting to note that succinate export flux mirrors the activity pattern of fumarase (*Figure 3*), potentially indicating that the Crabtree effect occurs only under conditions that result in succinate export. This suggests further investigation of this effect, perhaps through *in vitro* studies of worms exposed to different oxygen and glucose concentrations.

The glyoxylate pathway is primarily discussed in the literature as a way for plants to process fatty acids into glucose and starches, but it is also found in fungi, some protists, and bacteria (*Kondrashov et al., 2006*). The relevant genes have been detected in several Metazoan species, but the nematodes are the only group where the pathway is widely accepted to be functional, and where it is regarded as a conduit from fatty acids to glucose during embryogenesis. It involves many of the same enzymes as the TCA cycle but includes a 'shortcut' from isocitrate to succinate and malate. It has been suggested that this pathway may also play a role in energy metabolism (*Butler et al., 2012*)—in particular when metabolizing fatty acids—and our results appear to support that this hypothesis is plausible.

Besides predicting the use of metabolic pathways, our model has the ability to identify reactions essential to growth as potential therapeutic targets. Previous predictions of essentiality based on a

compartmentalized model of *C. elegans* metabolism, *iCEL1273*, were broadly consistent with *in vitro* gene essentiality screens (*Yilmaz and Walhout, 2016*), suggesting the process by which we made our predictions is likely to be biologically relevant.

Two of the highly prioritized drug target hits identified in our study were aldolase isozymes *aldo-1* (Bm5580) and *aldo-2* (Bm3135). Knocking out the cognate reactions led to unrecoverable states in the model, both genes have several druggable homologs in ChEMBL, and the expression of both genes was consistent across the adult life stages (*aldo-1* had the highest stage-specific expression out of any of the prioritized genes). These were not pursued in this work, as in a previous study we examined the effects of knocking down these genes with RNAi in adult female *B. malayi* (*Voronin et al., 2016*). While the knockdown of *aldo-1* had no significant effects, that of *aldo-2* showed effects including a decrease in the *Wolbachia* population, a reduction in fecundity of female worms, and an increase in apoptotic embryos. This shows that our model is capable of predicting viable targets in the adult worms.

We validated our predictions of essential reactions by observing the effects of three existing drugs on *B. malayi* adults. Fosmidomycin was originally investigated as an antibiotic in the 1980s, but more recently has been studied as an anti-malarial drug (*Umeda et al., 2011*; *Jomaa et al., 1999*; *Armstrong et al., 2015*). It acts on the non-mevalonate isoprenoid biosynthesis pathway, which is generally only found in some bacteria and plants, in addition to the apicoplast of *Plasmodium falciparum*. This helps to contribute to the drug's excellent safety profile. Tenofovir is currently licensed by the FDA for treatment of HIV and Hepatitis B (*Agarwal et al., 2015*). It is used as a nucleoside reverse transcriptase inhibitor, and its action against *B. malayi* may work by a similar mechanism, by competitive inhibition with AMP of adenylate kinase. MDL-29951 was identified as an inhibitor of fructose 1,6-bisphosphatase as a potential treatment for diabetes (*Wright et al., 2003*). Interestingly, this enzyme has recently and independently been proposed as a drug target against *Leishmania* species (*Yuan et al., 2017*). That study also solved several crystal structures of the enzyme, which would prove valuable for future refinements of the drug. There have also been several other inhibitors generated and tested against this enzyme from different species, which could provide a rich starting point for future work to refine the antiparasitic activity (*Dang et al., 2009*; *Dang et al., 2010*; *Tsukada et al., 2010*; *Kaur et al., 2017*). Both Tenofovir and MDL-29951 were expected to act against *B. malayi*, but primarily resulted in a significant reduction of *Wolbachia* populations when tested. The mechanism of action of these drugs against the endosymbiont is unclear, but *Wolbachia* does possess a homolog of both drug targets—(Wbm0321 (adenylate kinase) for Tenofivir; Wbm0158 (fructose-1,6-biphosphatase) for MDL-29951)—that the drugs may be acting against.

All three of the drugs tested in this study appear to possess activity against adult *B. malayi* worms, via reduction in *Wolbachia* populations and/or microfilaria production. The successful validation of these effects suggests that our metabolic model is a useful approximation of the worm. Consistent with metabolic reconstructions generated for other organisms such as Yeast and *E. coli*, we expect future iterations of our metabolic reconstruction will undergo further modifications involving, for example, the inclusion of new reactions or refinement of existing reactions as new experimental data is generated. Further research on this model may therefore yield even more therapeutic targets.

## Materials and methods

### Metabolic reconstruction and flux balance analyses

A draft metabolic reconstruction was generated as described previously (*Song et al., 2013*; *Blazejewski et al., 2015*; *Cotton et al., 2016*). Briefly, sets of metabolic enzymes were identified from the *B. malayi* gene models using DETECT V2 (*Hung et al., 2010*), BLASTP (*Camacho et al., 2009*) searches against enzymes curated in the SWISSPROT database (*Bateman and UniProt Consortium, 2015*), PRIAM (*Claudel-Renard et al., 2003*), EFICAz (*Tian et al., 2004*), and the BRENDA database (*Schomburg et al., 2002*). Assignment of metabolic pathways and gap-filling in the reconstruction were performed by Pathway Tools (*Karp et al., 2016*) and comparisons to our previously published *Onchocerca volvulus* metabolic reconstruction. As an example, if our reconstruction was missing one reaction out of a pathway that was complete in the *O. volvulus* network, that reaction

would be added to our model if reactions existed to consume the resulting metabolites. The network was then manually curated to ensure that the major pathways—such as the TCA cycle, fatty acid metabolism, and anaerobic metabolism—were complete and could be used by the model. The model was divided into three compartments representing the cytosol and mitochondria of *B. malayi*, as well as the *Wolbachia* endosymbiont. Where orthologous relationships could be inferred, *B. malayi* reactions were assigned to either the cytosol or mitochondrion on the basis of similar designations in the *i*CEL1273 metabolic model for *Caenorhabditis elegans* (*Yilmaz and Walhout, 2016*). Additional compartment assignments were performed with reference to literature sources. In the absence of such information, reactions were duplicated such that a cytosolic and a mitochondrial form were both available. Reaction reversibility is a very important determinant of model performance, and was primarily determined here by comparison to the previously published *O. volvulus* (*Cotton et al., 2016*) and *C. elegans* (*Yilmaz and Walhout, 2016*) models. Additional reactions that led to the formation of gasses such as $CO_2$ were also set to be irreversible (with the exception of known gas-consuming reactions such as the consumption of oxygen in oxidative phosphorylation and fatty acid catabolism).

Metabolites are confined to a single compartment and can only participate in reactions in other compartments if shuttled there by an explicit transport reaction. Most of these transporters were taken from the *i*CEL1273 model, except when contraindicated by *Berg, 2002*. Since *Wolbachia* lacks the genes necessary to produce NAD+, coenzyme A (CoA), ubiquinol, and folate (*Voronin et al., 2016*), reactions were added to allow the transport of each metabolite into the *Wolbachia* compartment from the cytosol. The integrity of the model was tested using FBA to ensure that it was able to produce biomass (as defined below), and the major metabolite fluxes were manually traced to ensure they were not the product of futile cycles or other biologically unfeasible artifacts.

All network analysis methods, including FBA, were performed using the cobrapy package version 0.5.10 (*Ebrahim et al., 2013*), which is a Python-based implementation of the popular COBRA toolbox of FBA-associated methods (*Schellenberger et al., 2011*). One drawback of FBA is that there are usually many different sets of reaction fluxes that can lead to the same solution, with no accepted way to choose the most biologically relevant. Parsimonious FBA (pFBA) is a permutation of this algorithm that first maximizes the objective function, and then identifies the solution with the smallest sum over all reaction fluxes. This is predicted to yield more biologically relevant solutions, and helps to ensure that analyses of the model under different conditions are likely to yield comparable sets of reaction fluxes.

## The biomass objective function

The overall biomass objective function is defined as

$$B = \delta \cdot (B_B + 0.1 \cdot B_W),$$

where $\delta$ is a positive scaling factor that can be used to calibrate the model output to equal some physical measurement, such as growth rate in bacteria; this becomes more challenging to interpret and measure in eukaryotes. The value has no effect on our results or conclusions but was set to 6.17336 so as to be consistent with our previous models. $B_B$ and $B_W$ are the *B. malayi*- and *Wolbachia*-specific biomass objective functions, and the coefficient on $B_W$ controls the *Wolbachia* load (see *Figure 2*). The specific biomass objective functions are defined as

$$B_B = 0.7089 \cdot AA + 0.2027 \cdot AA_M + 0.0198 \cdot RNA + 0.0169 \cdot Sat + 0.0132 \cdot Unsat + \\ 0.013 \cdot Misc + 0.0123 \cdot Lipids + 0.0082 \cdot Cof + 0.005 \cdot DNA$$

and

$$B_W = 0.4 \cdot AA_W + 0.25 \cdot RNA_W + 0.24 \cdot Lipids_W + 0.1 \cdot DNA_W + 0.01 \cdot Cof_W.$$

Here, variables with no subscript pertain to the *B. malayi* cytosol, those with an *M* subscript pertain to the *B. malayi* mitochondria, and those with a *W* subscript pertain to the *Wolbachia* cell. *AA* indicates the quantity of amino acids that have been expressed and assembled into proteins; *RNA* and *DNA* indicate the quantity of nucleotides that have been assembled into their respective nucleic acids; *Sat* and *Unsat* refer to the quantity of saturated and unsaturated fatty acids,

respectively; *Lipids* refers to the quantity of membrane-associated or other lipids; *Cof* refers to the quantity of cofactors; *Misc* and refers to the quantity of several miscellaneous compounds.

The polymer assembly reactions are defined as

$$AA + 0.35 \cdot ADP + 2.35 \cdot PO_4 + 2 \cdot GDP + 2.35 \cdot H^+ = AA^{RAW} + \\ 2.35 \cdot H_2O + 0.35 \cdot ATP + 2 \cdot GTP \cdot$$

for amino acids,

$$RNA + 0.35 \cdot ADP + 0.35 \cdot PO_4 + 0.35 \cdot H^+ = RNA^{RAW} + 0.35 \cdot H_2O + 0.35 \cdot ATP$$

for RNA, and

$$DNA + 0.35 \cdot ADP + 0.35 \cdot PO_4 + 0.35 \cdot H^+ = DNA^{RAW} + 0.35 \cdot H_2O + 0.35 \cdot ATP$$

for DNA.

The amino acid proportions making up each quantity of protein are defined by

$$AA^{RAW} = 0.066 \cdot Glu + 0.049 \cdot Gly + 0.06 \cdot Ala + 0.063 \cdot Lys + 0.054 \cdot Asp + 0.056 \cdot \\ Arg + 0.041 \cdot Gln + 0.082 \cdot Ser + 0.025 \cdot Met + 0.011 \cdot Trp + 0.044 \cdot Phe + 0.032 \cdot \\ Tyr + 0. \cdot 22 \cdot Cys + 0.093 \cdot Leu + 0.024 \cdot His + 0.042 \cdot Pro + 0.053 \cdot Asn + 0.057 \cdot \\ Val + 0.056 \cdot Thr + 0.069 \cdot Ile$$

and

$$AA_W^{RAW} = 0.068 \cdot Glu + 0.067 \cdot Gly + 0.065 \cdot Ala + 0.086 \cdot Lys + 0.053 \cdot Asp + 0.043 \cdot \\ Arg + 0.03 \cdot Gln + 0.077 \cdot Ser + 0.024 \cdot Met + 0.007 \cdot Trp + 0.042 \cdot Phe + 0.033 \cdot Tyr + \\ 0.013 \cdot Cys + 0.089 \cdot Leu + 0.019 \cdot His + 0.032 \cdot Pro + 0.053 \cdot Asn + 0.071 \cdot Val + \\ 0.044 \cdot Thr + 0.085 \cdot Ile.$$

The proportions making up each quantity of RNA and DNA are defined by

$$RNA^{RAW} = 0.324 \cdot ATP + 0.208 \cdot GTP + 0.178 \cdot CTP + 0.290 \cdot UTP,$$

$$RNA_W^{RAW} = 0.345 \cdot ATP + 0.214 \cdot GTP + 0.145 \cdot CTP + 0.295 \cdot UTP,$$

$$DNA^{RAW} = 0.354 \cdot dATP + 0.146 \cdot dGTP + 0.146 \cdot dCTP + 0.354 \cdot dTTP,$$

$$DNA_W^{RAW} = 0.329 \cdot dATP + 0.171 \cdot dGTP + 0.171 \cdot dCTP + 0.329 \cdot dTTP.$$

The remaining biomass categories are defined by

$$Sat = 0.392 \cdot Hex + 0.42 \cdot Oct + 0.1 \cdot Dod + 0.088 \cdot Tet,$$

where the compounds are hexadecanoic acid, octadecanoic acid, dodecanoic acid, and tetradecanoic acid, respectively;

$$Unsat = 0.084 \cdot Ara + 0.369 \cdot Ole + 0.547 \cdot Lin,$$

where the compounds are arachidonate, oleic acid, and linoleate, respectively;

$$Lipids = 0.453 \cdot Pch + 0.073 \cdot Cho + 0.314 \cdot Pet + 0.078 \cdot Sph + 0.039 \cdot Pmy + 0.043 \cdot Pse,$$

where the compounds are phosphatidylcholine, cholesterol, phosphatidylethanolamine, sphingomyelin, 1-phosphatidyl-D-myo-inositol, and phosphatidylserine, respectively;

$$Cof = 0.125 \cdot NAD^+ + 0.125 \cdot NADP^+ + 0.125 \cdot CoA + 0.125 \cdot FAD + 0.125 \cdot Pyp + \\ 0.125 \cdot Heme + 0.125 \cdot Uag + 0.125 \cdot TFH,$$

where the compounds are nicotinamide adenine dinucleotide, nicotinamide adenine dinucleotide phosphate, coenzyme A, flavin adenine dinucleotide, pyridoxal phosphate, heme, UDP-N-acetyl-alpha-D-glucosamine, and tetrahydrofolate, respectively;

$$Misc = 0.36 \cdot Spr + 0.079 \cdot Ggp + 0.561 \cdot Spn,$$

where the compounds are spermidine, geranylgeranyl diphosphate, and spermine, respectively.

## Generation of transcriptomic data for the *Wolbachia* endosymbiont during molting

Previously published transcriptome data were used for this study, with the exception of the *Wolbachia* trascriptome during the molting of the worm from L3, L3 day 6, and L3 day 9. These data were generated as described in *Grote et al., 2020*. In brief, all *B. malayi* worms were obtained from FR3 (Filariasis Research Reagent Resource Center; BEI Resources, Manassas, VA, USA). Infective third-stage larvae (iL3) were recovered from mosquitoes (*Aedes aegypti*) and mammalian stage larvae were recovered from gerbils (*Meriones unguiculatus*) at 6 and 9 days post infection (dpi). Total RNA was prepared from *B. malayi* worms and *Wolbachia* as previously described (*Grote et al., 2017*). RNA was prepared from three biological replicates of infective L3 (iL3; 2000 larvae each), 3 replicates of 6 dpi larvae (1500 each) and 2 replicates of 9 dpi larvae (1300 each). Libraries were prepared using the NEBNext Ultra II RNA Library Prep Kit for Illumina (New England Biolabs) according to manufacturer instructions. Libraries were sequenced at NYU's GenCore on the Illumina Next-Seq500 platform with 150 bp paired-end reads. Sequence reads from each sample were analyzed with the Tuxedo suite of tools (*Trapnell et al., 2010*; *Trapnell et al., 2013*; *Kim et al., 2013*). Reads were mapped with Tophat2's Bowtie2-very-sensitive algorithm to the genome assembly of *Wolbachia* of *B. malayi*(*Foster et al., 2005*). Each biological replicate received an average of 1.3 million reads that mapped to the *Wolbachia* genome. The resulting BAM files were then used with HtSeq to obtain raw read counts. Differential gene expression analysis was performed using EdgeR (*Robinson et al., 2010*), Data were combined with previously published stages 16 dpi (L4), and male and female worms at 30, 42 and 120 dpi (*Grote et al., 2017*).

## Generation of life stage specific metabolic models

To understand how metabolic pathway dependencies may vary across the *B. malayi* life cycle, we integrated new and existing stage-specific RNA-Seq datasets to generate life stage specific metabolic models. Of the 849 enzymatic reactions in the model with gene evidence, 837 had stage-specific expression data, allowing constraints to be placed on their associated metabolic flux. RNA-Seq enzyme expression was used to apply constraints on reaction flux as we have done previously (*Song et al., 2013*). In brief, the expression of each gene was normalized across life stages, and the relative expression for each life stage was applied to the upper and lower bounds of all associated reactions. For example, if the expression of a gene in one particular life stage was measured to be 30% of its maximum across all life stages, the corresponding lower and upper reaction bounds for that life stage model would be set to (0, 300) or (−300, 300) in arbitrary flux units, for irreversible and reversible reactions, respectively; the default bounds are (0, 1000) or (−1000, 1000) for irreversible and reversible reactions, respectively. As we cannot determine whether a measured expression of zero indicates no expression or that the transcripts were simply not sequenced, all reactions with a measured expression of zero were left unconstrained in the model. This yielded 11 distinct metabolic models: open or unconstrained (i.e. without any expression constraints), L3, L3D6, L3D9, L4, F30, F42, F120, M30, M42, and M120; where L3D6 indicates third-stage larvae at 6 days post-infection, and F30 and M30 indicate adult female and male worms 30 days post-infection, respectively.

## Metabolomics sample preparation and run

All parasites were obtained from FR3 (Filariasis Research Reagent Resource Center; BEI Resources, Manassas, VA, USA) where they were isolated and separated by sex from infected gerbils (*Meriones unguiculatus*) or mosquitoes (*Aedes aegypti*). Worms were flash-frozen and shipped to the New York Blood Center for processing. Stages used for metabolomics analysis included L3 larvae from mosquitoes, adult male and female worms at 120 dpi, and microfilaria. The number of worms per sample were 20 adult female worms, 40 adult males, $2 \times 10^6$ microfilariae, and 200 L3 larvae per biological replicate. Samples were washed in 1x PBS and run in triplicate. Adult male and female worms were picked individually from PBS and each biological was weighed. The microfilaria and L3 samples were spun down, the PBS pipetted off, and weighed directly into a metabolomics 2 mL screw cap vial

with total amounts ranging from 1.3 mg (adult males) to 15.8 mg (microfilaria). Metabolites were extracted and the data analyzed as described below.

## Metabolomics extraction and analysis

### Metabolite extraction

The mass of the weighed worm samples was used to scale the metabolite extraction to a ratio of 16.5 mg / 1 mL extraction solvent. Freezing 80% acetonitrile was added directly to each vial containing the samples, along with zirconium disruption beads (0.5 mm, RPI) and homogenized for 3 min at 4°C in a BeadBlasterTM with a 30 s on, 30 s off pattern. The resulting lysate was centrifuged at 21,000 x g for 3 min, and 90% of the supernatant volume was transferred to a 1.5 mL microfuge tube for speed vacuum concentration, no heating. The dry extracts were resolublized in a volume of LCMS grade water 1/10th of that used for the homogenization step, sonicated in a water bath for 3 min, and transferred to a glass insert for analysis.

### Data acquisition

All data were acquired by liquid chromatography coupled to high resolution tandem mass spectrometry (LC-MS/MS). The LC column was a MilliporeTM ZIC-pHILIC (2.1 × 150 mm, 5 μm) coupled to a Dionex Ultimate 3000TM system and the column oven temperature was set to 25°C for the gradient elution. A flow rate of 100 μL/min was used with the following buffers; A) 10 mM ammonium carbonate in water, pH 9.0, and B) neat acetonitrile. The gradient profile was as follows; 80–20%B (0–30 min), 20–80%B (30–31 min), 80–80%B (31–42 min). Injection volume was set to 2 μL for all analyses (42 min total run time per injection). MS analyses were carried out by coupling the LC system to a Thermo Q Exactive HFTM mass spectrometer operating in heated electrospray ionization mode (HESI). Method duration was 30 min with a polarity switching data-dependent Top five method for both positive and negative modes. Spray voltage for both positive and negative modes was 3.5kV and capillary temperature was set to 320°C with a sheath gas rate of 35, aux gas of 10, and max spray current of 100 μA. The full MS scan for both polarities utilized 120,000 resolution with an AGC target of 3e6 and a maximum IT of 100 ms, and the scan range was from 67 to 1000 m/z. Tandem MS spectra for both positive and negative mode used a resolution of 15,000, AGC target of 1e5, maximum IT of 50 ms, isolation window of 0.4 m/z, isolation offset of 0.1 m/z, fixed first mass of 50 m/z, and 3-way multiplexed normalized collision energies (nCE) of 10, 35, 80. The minimum AGC target was 1e4 with an intensity threshold of 2e5. All data were acquired in profile mode.

### Data analysis

Metabolomics data were processed with two approaches called the hybrid analysis and predictive analysis. The resulting relative metabolite intensities for both methods were then processed with an in-house pipeline for statistical analyses and plot generation using a variety of custom Python code and R libraries including: pheatmap, MetaboAnalystR, manhattanly. For the hybrid analysis, peak height intensities were extracted based on the established accurate mass and retention time for each metabolite as adapted from the Whitehead Institute (*Chen et al., 2016*), and verified with authentic standards and/or high resolution MS/MS manually curated against the NIST14MS/MS (*Simón-Manso et al., 2013*) and METLIN (*Smith et al., 2005*) spectral libraries. The theoretical m/z of the metabolite molecular ion was used with a ± 10 ppm mass tolerance window, and a ± 0.2 min peak apex retention time tolerance within the expected elution window (1–2 min). The median mass accuracy vs the theoretical m/z for the library was −4.3 ppm (n = 127 detected metabolites). Median retention time range (time between earliest and latest eluting sample for a given metabolite) was 0.24 min (30 min LCMS method). A signal to noise ratio (S/N) of 3X was used compared to blank controls throughout the sequence to report detection, with a floor of 10,000 (arbitrary units). For the predictive analysis, metabolites were detected and quantified by matching the predicted m/z of the metabolite as follows. All predicted model metabolites (n = 865) were filtered to the subset with HMDB ID and a neutral chemical formula (n = 553). The formula of each metabolite was then used to predict both the positive and negative mode molecular ions excluding adducts, multiply charged species, or more complex ion types that is [M+H]+ and [M-H]- (n = 1106 total m/z values for extraction). A ± 10 ppm mass tolerance window was again used to extract the peak intensities for all m/z values with a ± 0.2 min peak apex retention time tolerance, centering the window on the retention

time of the highest intensity peak for each metabolite across all samples. This approach does not discriminate between isomers, for instance 95 of the 553 metabolites in the predicted dataset shared a chemical formula with another metabolite, but in many cases the isomerization was due to stereochemistry (e.g., R-lactic acid vs S-lactic acid) while in other cases formula isomers could confound the data interpretation (dGMP vs AMP). The resulting putative metabolite peak heights were further filtered as follows, after applying the 3X S/N threshold. Metabolites were limited to those that were detected in at least three replicates (n = 3 per group). Next, an average intensity was calculated for each metabolite across all samples and the data were sorted from high intensity to low intensity. Duplicate metabolite names were removed, thereby keeping the row result from whichever polarity gave a stronger signal (e.g., glutamine was detected in both polarities at 11.5 min, but gave a stronger signal in positive mode).

## Validation of drug targets

For validation, 15 adult female and 15 adult male worms (120 dpi) were cultured *in vitro* for each drug treatment or control group. Worms were cultured for six days in a 12-well plate with two worms per well, in complete culture medium (RPMI-1640 supplemented with 10% FBS, 100 U/mL penicillin, 100 mg/mL streptomycin, 2 mM L-glutamine) at 37°C under 5% $CO_2$. Three drugs were assayed: Fosmidomycin, MDL-29951, and Tenofovir, each at a concentration of 12 µM. Media was changed every other day.

Microfilarial release by female worms was determined by quantifying the number present in the media on the 6th day of treatment and collection of adult worms. Two technical replicates and 10 biological replicates were used to determine microfilaria production. Averages are presented with their standard deviations. Significance was determined by a single factor ANOVA, followed by t-tests between each treatment group and the control using a Bonferroni correction.

To analyze the number of *Wolbachia* per worm, DNA was extracted from adult male worms using the QIAmp DNA Mini Kit (QIAGEN) according to the manufacturer's protocol. *Wolbachia* per individual worm was calculated by genomic qPCR using primers for a single-copy *Wolbachia* gene (*wsp*; accession AAW71020). Each treatment or control group had 8 (or 9) biological replicates, and each replicate contained three male worms. Averages are presented with their standard deviations. Significance was determined by a single factor ANOVA, followed by t-tests between each treatment group and the control using a Bonferroni correction.

The Worminator system (*Storey et al., 2014*) was used to assess changes in motility upon drug treatment. Motility of adult female worms was assessed using one female worm per well in a 12 well plate. Each treatment or control had eight biological replicates.

## Data availability

All of our metabolic models–the unconstrained and life stage-specific variants–are freely available at https://github.com/ParkinsonLab/Brugia_metabolic_network.

Our transcriptomics data for the *B. malayi* molt (life stages L3 to L3D6 to L3D9) are available through the Sequence Read Archive (PRJNA557263).

All of our metabolomics data – from both the hybrid and predictive analyses – are freely available at https://github.com/ParkinsonLab/Brugia_metabolic_network (*Curran et al., 2020*; copy archived at https://github.com/elifesciences-publications/Brugia_metabolic_network).

## Acknowledgements

This research was supported by a grant to JP and EG from the National Institutes of Health (R21AI126466). Additional sources of funding include the Natural Sciences and Engineering Research Council to JP (NSERC; RGPIN-2014–06664). NN was supported by a student Restracomp fellowship given by the Hospital for Sick Children. Funding for AG was provided by the T32 Ruth L Kirschstein Institutional National Research Service Award (T32AI007180) and the F31 Ruth L Kirschstein Pre-doctoral Individual NRSA (F31AI131527). Worms for this study were provided by FR3 (Filariasis Research Reagent Resource Center; BEI Resources, Manassas, VA, USA). New York University's Genomics Core (GenCore) is supported by the Zegar Family Foundation.

## Additional information

### Funding

| Funder | Grant reference number | Author |
| --- | --- | --- |
| National Institutes of Health | R21AI126466 | Elodie Ghedin<br>John Parkinson |
| Natural Sciences and Engineering Research Council of Canada | RGPIN-2014-06664 | John Parkinson |
| Hospital for Sick Children Research Training Centre | | Nirvana Nursimulu |
| Ruth L Kirschstein Institutional National Research Service | T32AI007180 | Alexandra Grote |
| Ruth L Kirschstein Pre-doctoral Individual NRSA | F31AI131527 | Alexandra Grote |

The funders had no role in study design, data collection and interpretation, or the decision to submit the work for publication.

### Author contributions

David M Curran, Conceptualization, Data curation, Software, Formal analysis, Investigation, Visualization, Methodology; Alexandra Grote, Conceptualization, Resources, Formal analysis, Validation, Investigation, Methodology; Nirvana Nursimulu, Data curation, Visualization, Methodology; Adam Geber, Resources, Methodology; Dennis Voronin, Resources, Validation, Investigation, Methodology; Drew R Jones, Resources, Formal analysis, Validation, Investigation, Methodology; Elodie Ghedin, Conceptualization, Supervision, Funding acquisition, Validation, Investigation, Project administration; John Parkinson, Conceptualization, Supervision, Funding acquisition, Investigation, Visualization, Project administration

### Author ORCIDs

David M Curran https://orcid.org/0000-0003-2749-869X
Adam Geber http://orcid.org/0000-0003-3022-0525
Drew R Jones http://orcid.org/0000-0001-8732-9818
Elodie Ghedin https://orcid.org/0000-0002-1515-725X
John Parkinson https://orcid.org/0000-0001-9815-1189

### Decision letter and Author response

Decision letter https://doi.org/10.7554/eLife.51850.sa1
Author response https://doi.org/10.7554/eLife.51850.sa2

## Additional files

### Supplementary files

• Supplementary file 1. Supplemental information.
• Supplementary file 2. Metabolomics.
• Supplementary file 3. Drug target hits.
• Transparent reporting form

### Data availability

All of our metabolic models-the unconstrained and life stage-specific variants-are freely available at https://github.com/ParkinsonLab/Brugia_metabolic_network. Our transcriptomics data for the B. malayi molt (life stages L3 to L3D6 to L3D9) are available through the Sequence Read Archive (PRJNA557263). All of our metabolomics data - from both the hybrid and predictive analyses - are

freely available at https://github.com/ParkinsonLab/Brugia_metabolic_network (copy archived at https://github.com/elifesciences-publications/Brugia_metabolic_network).

The following previously published dataset was used:

| Author(s) | Year | Dataset title | Dataset URL | Database and Identifier |
|---|---|---|---|---|
| Grote A, Li Y, Liu C, Voronin D, Geber A, Lustigman S, Unnasch T, Welch L, Elodie G | 2020 | Genome-wide prediction and functional validation of transcription factor binding motifs in the parasitic nematode Brugia malayi | https://www.ncbi.nlm.nih.gov/bioproject/PRJNA557263 | NCBI BioProject, PRJNA557263 |

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
