## [Decision Letter]

**Acceptance summary:**

The manuscript employs a metabolic model of the filarial nematode *Brugia malayi* to predict metabolic pathways used by the parasite at different lifecycle stages and within different environments. The work represents one of the first comprehensive investigations using metabolic modelling to predict essential reactions in a parasitic nematode and offers a potentially useful strategy for the development of new drugs targeting these important pathogens. The authors validate their work by predicting several druggable targets based on the model, and proceed to validate three of these reactions using already available drugs.

**Decision letter after peer review:**

[Editors’ note: the authors submitted for reconsideration following the decision after peer review. What follows is the decision letter after the first round of review.]

Thank you for submitting your work entitled "Modeling the metabolic interplay between a parasitic worm and its bacterial endosymbiont identifies novel drug targets" for consideration by *eLife*. Your article has been reviewed by three peer reviewers, including Nicola L Harris as the Reviewing Editor and Reviewer #1, and the evaluation has been overseen by a Senior Editor. The following individual involved in review of your submission has agreed to reveal their identity: James B Lok (Reviewer #3).

Our decision has been reached after consultation between the reviewers. Based on these discussions and the individual reviews below, we regret to inform you that your work will not be considered further for publication in *eLife*.

Although reviewers agree the idea of using metabolic modelling approaches to identify novel drug targets for fillarial infection is of high interest, a number of deficiencies in the experimental rationale, modelling approach employed and communication of the rationale and outcomes were identified. In particular it was not clear why the authors did not choose the most recent work regarding *C. elegans* metabolic Reconstruction (WormJam: A consensus *C. elegans* Metabolic Reconstruction and Metabolomics Community and Workshop Series. Worm. 2017; 6(2): e1373939) The modelling of host conditions or the bacterium *Wolbachia* was felt to be insufficient in the current form.

Reviewer #1:

The manuscripts by Curran et al. employs a metabolic model of the filarial nematode *Brugia malayi* to predict metabolic pathways used by the worm at different lifecycle stages and within different environments. The authors then use the model to predict essential reactions useful for drug targeting and validated three of these reactions using already available drugs.

The work is both interesting and novel in that 1) new drug targets for Brugia are needed and valuable and 2) large scale metabolic modelling of a parasitic nematode has not been reported previously.

However as it stands a large amount of data is given and the rationale behind the experimental design is not always clear, making it difficult for the reader to determine how meaningful each dataset is. The manuscript could be greatly improved if more effort was made to restrict the data shown in the main figures related to environmental conditions (HOLG, LOLG, LOHG AND HOHG) to the those in that each lifecycle stage would be expected to actually encounter within the host. The relationship of each dataset to one another also needs to be commented on more clearly (for instance what environmental conditions were the predictions made in Figure 7 performed under?).

One of the most important figures in the manuscript is Figure 8 in which the ability of the model to predict drug targets was determined using adult worms. Given that the preceding sections of the manuscript focused heavily on determining the environmental factors influencing the model it would important to determine whether these same factors influence in the activity of the drugs. This could be achieved by performing the assays in cultures containing medium with different levels of glucose and under low or high oxygen conditions. This is particularly important given emerging data indicating that drug screening for anti-helminthics yields different outcomes when drugs are tested under high oxygen conditions (such as that employed in Figure 8) versus low oxygen conditions). Drugs should also be compared side by side in these assays to an “in use” anti-helminthic such as ivermectin.

Reviewer #2:

The paper presents the metabolic network model of a parasitic worm and its bacterial endosymbiont, and it uses this model to study the metabolic adaptations that allow the survival of the parasite. Metabolic models can be indispensable for the study of such complex systems, the interpretation of experimental data and the identification of drug targets.

The modeling effort presented in this manuscript suffers from major issues which make the present model(s) and their presentation unacceptable for final publication. When one uses a metabolic model, and any model, it is of fundamental importance to first assess the quality of modeling framework, before using the model for the analysis of experimental data. Otherwise, the results of the model-based analysis of the data are simply artifacts of the errors in the low-quality models.

The following are some of the major issues in the development and quality of the models in the manuscript.

* While many metabolites and nutrients are present in the worm cytosol, not all of them are available to bacterium. The conditional availability of the nutrients and metabolites will impact the conclusions about the physiology (and gene essentiality) of the bacterium.

* The model appears to be built based on a semi-manual curation. There are currently established methods and workflows that should be used (such as RAVEN). The authors should consider all these, discuss their capabilities and provide a justification why they are not using one of them. I think that couple of these methods have been optimized and any other procedure such the one followed here is at the best suboptimal.

* Within the established metabolic modeling workflows there is also the concept of "metabolic tasks" which defines the metabolic capabilities of the organisms and the ability of the model to fulfill these tasks. The authors must follow such workflow, define the metabolic tasks and assess the performance of their models with respect to these tasks. Otherwise, we cannot systematically evaluate the quality of the models.

* The model is using only two compartments of the host organism. This is already a major inefficiency of the models here. For every eukaryote that has been model with high quality additional other compartments have been considered (such as ER and nucleus) and their consideration has been shown to be important for capturing the physiology of the organism.

* It is not clear what is the information used for modeling transporters (Berg et al., 2002; not a proper citation and I have difficulty to find this with such limited and imprecise information). It is important to carefully consider which transporters across compartments and across cytosol and the bacterium are feasible. Careful consideration, curation, modeling and justification of the transporters is extremely important for the quality and the performance of the model.

* The authors ignore the most important "compartment" which is the environment of the human host. It has been shown in the past that modeling and properly contraining these interactions are essential for the evaluation of the models of parasites and on the use of these models for the analysis and interpretations of experimental data (e.g., PLoS Comput Biol. 2017 Mar; 13(3): e1005397 and PLoS Comput Biol. 2017 Mar; 13(3): e1005397.) For example the studies here should first identify what are the essential nutrients and what are the commonly and most probably used nutrients. The same applies also for modeling the nutrients available to bacterium (as mentioned earlier.)

* The authors use as a reference model the previously published model iCEL1273 for the worm E. elegans. However, the quality of this model is low and there is a later work for a consensus *C. elegans* metabolic Reconstruction (WormJam: A consensus *C. elegans* Metabolic Reconstruction and Metabolomics Community and Workshop Series. Worm. 2017; 6(2): e1373939) which should be consider. The authors should use this work as reference or provide a strong justification why they choose exclusive use iCEL1273, which is not of good quality.

* There is not a good analysis on the modeling of the bacterium Wolbachia. The choice of reactions is not justified, and again, there is not a clear procedure outlined for the reconstruction and modeling of the bacterium metabolic modeling.

* It also appears that important information on model development and statistics is missing. The only Supplementary File is an excel file with "Single knockout" data.

* The authors do not discuss the reversibility of the reactions and their thermodynamic feasibility. This is an essential procedure in model reconstruction and development, and it is well known and appreciated fact that reaction directionality has a significant impact in the performance of the model for flux estimation and gene essentiality analysis.

* The model simulations do not really offer a lot in our understanding of the physiology of the organisms. The authors could try to extract the import lessons from the simulations and communicate them in a more clear and informative way. However, and without a proper constraining and classification of the nutrient uptake fluxes, it is not possible to gain a clear understanding of the results presented here.

* It is very hard to assign significance in the essentiality analysis since the model reconstruction and the assignment of constraints has not been done properly and it has been justified in a convincing manner (as discussed above).

* The predictions discussed in the manuscript do not provide a sufficient validation of the model. Some of them could have been predicted without the use of the model. The small number of predictions does not provide enough evidence and confidence to support the claim the "metabolic model (in this manuscript) is a useful approximation of the worm”.

* It is not clear what are the methods used by the authors for the integration of transcriptomics and metabolomics data. There are exist many well established methods that are used for such integration and such method-based integration is the current standard in model reconstruction and analysis.

Reviewer #3:

This paper reports the development of a new computational model of intermediary metabolism in the lymphatic dwelling filarial nematode *Brugia malayi*. This model predicts metabolic flux through catabolic and biosynthetic pathways under varying conditions of oxygen tension and glucose availability in key life cycle stages of the parasite that infect the definitive host (post-infective L3, L4, Male and Females).

The paper has many strengths of a substantive nature, which are enumerated below. The paper is also very well written, and only a few minor changes are recommended in the text. The authors should give some attention to specific points about the presentation of Figures 1 and 2.

Substantive strengths

1) The metabolic model builds on others developed for *Onchocerca* and *C. elegans*. This is a logical approach and one that undoubtedly streamlined construction of the Brugia model.

2) Confidence in the fidelity of the model is engendered because it captures salient features of nematode metabolism, such as the switching between aerobic and anaerobic pathways, a predicted Crabtree effect under high oxygen and glucose, and use of glutamate to aspartate conversion to generate energy.

3) Great that the model predicts ability of the parasite to utilize aerobic and anerobic metabolism as it proceeds through various body compartments in its migration within the host.

4) The model prediction that glucose and oxygen are limiting factors in fitness of adult *B. malayi* is an interesting and worthwhile finding that could direct discovery of new compounds acting on energy metabolism.

5) Identification of scenarios where multiple redundant pathways are available to the parasite to achieve the same metabolic output also seems a very astute finding, as such pathways would presumably be less advantageous likely drug targets. ie. they are less likely to be essential.

6) Concordance between model predictions and actual metabolomics on extracts of staged parasites bolsters confidence in the model.

7) Through a process of in silico knockouts, the authors have predicted 99 enzymatic reactions that are essential. This demonstrates a crucial practical application of the model.

8) The authors definition of essential reactions as being ones whose disruption would

reduce predicted parasite biomass by 50% or more seems valid to this reviewer given that such a reduction in biomass would likely compromise the worms' ability to survive in the hostile environment of an immunocompetent host.

9) The "two hit" discovery from model simulations that two different reactions participate in pathways that lead to the same metabolite could facilitate the design of combination therapies that would greatly decrease the likelihood of drug resistance.

10) The authors used a rational and very practical scheme for identifying potential target pathways from among the 99 essential reactions predicted by the model. Criteria in this scheme included expression across multiple life stages, number and availability of existing inhibitors in the ChEMBL database and similarity to human homologs (presumably a detracting factor). Selected compounds from this list, targeting diverse pathways such as isoprenoid precursor biosynthesis, gluconeogenesis and purine metabolism, namely fosmidomycin, mdl-29951 and tenofovir, respectively, either decreased *Wolbachia* loads, and/or fecundity in cultured adult *B. malayi*. These results bolster confidence in the model's utility in identifying drug targets in Brugia spp, and perhaps in Wuchereria bancrofti as well.

[Editors’ note: further revisions were suggested prior to acceptance, as described below.]

Thank you for choosing to send your work entitled "Modeling the metabolic interplay between a parasitic worm and its bacterial endosymbiont identifies novel drug targets" for consideration at *eLife*. Your letter of appeal has been considered by a Senior Editor and a Reviewing editor, and we are prepared to consider a revised submission with no guarantees of acceptance.

Summary

The manuscripts employs a metabolic model of the filarial nematode *Brugia malayi* to predict metabolic pathways used by the worm at different lifecycle stages and within different environments. The authors then use the model to predict essential reactions useful for drug targeting and validated three of these reactions using already available drugs.

The work is both interesting and novel in large scale metabolic modelling of a parasitic nematode has not been reported previously and the reviewers agree that it offers information will be useful for the community, keeping in mind some major challenges when working with parasitic nematodes.

However it was felt that the demonstration of anti-filarial activity as shown for the three drugs is not sufficient validation of a genome scale model. The authors should detail under which model assumptions these validations will fail and state these as a calibration rather than a validation. Further validation should be provided as outlined under essential revisions below.

Essential revisions:

1) A combination of the following things should be provided as further validation of the model:

a) Predicted drugs being effective. At the current time it isn't possible to judge whether this is significant in absence of the knowledge regarding how the drugs tested were selected from the prioritized list? Details of how the three drugs were selected from the prioritized list should be provided (Presumably, the mentioned targets were not the only targets associated with these drugs in Chembl? If so, maybe listing the other associated targets in a supplement would be useful, since it is possible that it is some of those targets that may be involved in the activities observed)

b) Gene essentiality comparison with *C. elegans*. A proper comparison should be made inclusive of statistics. If the model predictions have significant enrichment of such genes, this would be helpful.

c) If there is data out there about *wolbachia* load in different stages, this could be another validation if it is consistent with what the model predicts in terms of optimal *wolbachia* load under different stage transcriptome-based constraints.

2) A few more cases of FBA's (and pFBA's) demonstrated utility in uncovering real biologically relevant insights should be included in the Introduction along with corresponding references.

---

## [Author Response]

[Editors’ note: The authors appealed the original decision. What follows is the authors’ response to the first round of review.]

Although reviewers agree the idea of using metabolic modelling approaches to identify novel drug targets for fillarial infection is of high interest, a number of deficiencies in the experimental rationale, modelling approach employed and communication of the rationale and outcomes were identified. In particular it was not clear why the authors did not choose the most recent work regarding C. elegans metabolic Reconstruction (WormJam: A consensus *C. elegans* Metabolic Reconstruction and Metabolomics Community and Workshop Series. Worm. 2017; 6(2): e1373939) The modelling of host conditions or the bacterium Wolbachia was felt to be insufficient in the current form.

We would first like to thank you for taking the time to oversee the reviews of our manuscript. While we were obviously disappointed with your initial decision, in reading over the reviews, we became concerned that the decision may have been unduly influenced by the comments provided by reviewer 2 which (as we highlight below) we consider to be well off base for current expectations in the field. We note that the other two reviewers were very positive about the study, highlighting, in the case of reviewer 3, ten substantive strengths.

As you can appreciate, despite its global health importance, *Brugia malayi* remains an extremely challenging organism to work with. Consequently, unlike model organisms, or the experimentally tractable Plasmodium parasite, biochemical data for Brugia is greatly limited, an issue that perhaps is not apparent to reviewer 2. By attempting to crystalize current knowledge of Brugia metabolism, this study provides a valuable resource for the community that has been demonstrated to generate new hypotheses, including drug targets that we have successfully validated. We therefore politely suggest that the points raised by reviewer 2 do not reflect the current state of the field and, as we show below, we believe that each of their comments are readily addressable. We are therefore writing to request whether you would be willing to consider a revised version of our study which addresses each and every one of the comments raised by the reviewers.

Reviewer 2 expressed the following concerns:

1) It is not clear which metabolites are available to the worm and its endosymbiont which impact predictions

This is a common criticism of any metabolic model and reflects an incomplete knowledge of transport reactions between cellular compartments, particularly for non-model organisms. Such reactions are challenging to infer computationally and typically identified through biochemical studies. Here we chose to be conservative and included transport reactions as required by reactions assigned to the compartments. This is a standard approach used in the field has the potential to result in false negatives, but importantly, will not impact false positives. Hence any essential reactions we predict in such a model will remain essential irrespective of additional transport constraints.

2) The authors do not use established methods and workflows such as RAVEN or concepts such as metabolic tasks

We would politely point out that, as outlined by Machado et al. NAR 2018, our approach is equivalent to the many tools that use a bottom-up approach including: (i) annotate genes with metabolic functions; (ii) retrieve the respective biochemical reactions from a reaction database, such as KEGG (26); (iii) assemble a draft metabolic network; (iv) manually curate the draft model. Our approach has been applied to several reconstructions including: Song et al., 2013; Blazejewski et al., 2015; Cotton et al., 2016; International Helminths Genome Consortium Nature Genetics 2019). We would add that a key advance of our approach over other approaches is the use of more sensitive enzyme annotation tools to complete the first step. We can of course provide additional details concerning the construction of our model.

3) Unlike other eukaryotic models, only two host compartments are considered in the Brugia model and that, unlike other studies of parasites, the human host is ignored.

As noted above, Brugia is a challenging organism to work with and lacks the depth of knowledge available for other organisms to accurately assign metabolic reactions to specific compartments. We further note that the *C. elegans* model (iCEL1273) also features only two compartments. Regarding the comment concerning the human host, the reviewer cites a study (actually they cited the same study twice) to illustrate how additional data is used to improve a previously published model of Plasmodium metabolism. In the same way, we would expect that our initial model will be refined as new data becomes available.

4) The model simulations do not offer a lot of our understanding of the physiology of the organisms

This seems to be entirely contradicted by the ten substantive strengths outlined by reviewer 3 who appears to be more knowledgeable concerning Brugia physiology.

5) We compare our model to iCEL1273 which is deemed low quality and should compare to a later work

We were puzzled by this comment as the reviewer does not provide any evidence to support their claim that the quality of iCEL1273 is low. We were also initially puzzled by the suggestion to use the reconstruction referred to in WormJam: A consensus *C. elegans* Metabolic Reconstruction and Metabolomics Community and Workshop Series. Worm. 2017; 6(2): e1373939. This latter reference appears to report on a community effort to develop a metabolic reconstruction for *C. elegans*, but provides no link to any actual reconstruction. After some digging we found Witting et al. Frontiers in Molecular Biosciences 2018 which describes the integration of four previously published models including iCEL1273. We can of course compare our model to this new model in any revision. But perhaps more importantly this illustrates exactly the process of how initial metabolic reconstructions can be further embellished as new data comes in. However, this process cannot work if the initial models do not get published!

6) It is hard to assign significance to the essentiality analysis

As noted above, our model is conservative and we expect that we may have missed some essential reactions. Nonetheless we stand by our predictions of essential reactions: due to limitations in obtaining worms for these experiments, we were only able to target 3 reactions; satisfyingly we validated all three. This raises a key point, due challenges in obtaining worms for such experiments, resources such as the one presented are critical to help prioritize experiments for in vitroexperiments that rely on such a rare resource.

[Editors’ note: what follows is the authors’ response to the second round of review.]

Essential revisions:1) A combination of the following things should be provided as further validation of the model:a) Predicted drugs being effective. At the current time it isn't possible to judge whether this is significant in absence of the knowledge regarding how the drugs tested were selected from the prioritized list? Details of how the three drugs were selected from the prioritized list should be provided (Presumably, the mentioned targets were not the only targets associated with these drugs in Chembl? If so, maybe listing the other associated targets in a supplement would be useful, since it is possible that it is some of those targets that may be involved in the activities observed)

As requested we now provide more details concerning the prioritization strategy together with a full table (Subsection “Fosmidomycin, MDL-29951, and Tenofovir possess antifilarial activity” and Supplementary file 3).

“To validate the performance of our model, we selected a subset of reactions for targeted inhibition using known drugs. Of the 102 reactions predicted to be essential, 77 were associated with one or more genes (33 in the cytosol, 41 in Wolbachia, and 3 in the mitochondria). This subset was chosen because they were considered less likely to be model artifacts. Reactions were prioritized by considering their expression across different life stages, the number of inhibitors identified in the ChEMBL database (Gaulton et al., 2012; Davies et al., 2015; Gaulton et al., 2017), and the similarity to human homologs (see Supplementary file 2 for details). From this list we selected three inhibitors to validate our predictions through in vitro assays (Table 1), primarily based on their cost and availability from suppliers.”

b) Gene essentiality comparison with *C. elegans*. A proper comparison should be made inclusive of statistics. If the model predictions have significant enrichment of such genes, this would be helpful.

As recommended we performed a one-tailed Fisher's Exact Test comparing these sets, and calculated a p-value of 1.9E-11 (meaning the probability of finding 52 or more hits out of 71 choices by random chance is 1.9E-11).

“73% of the predicted iDC625 essential reactions overlap with the experimentally determined essential reactions of *C. elegans* (significance determined by one-tailed Fisher’s Exact Test; hypergeometric pvalue = 1.9E^-11^).”

c) If there is data out there about wolbachia load in different stages, this could be another validation if it is consistent with what the model predicts in terms of optimal wolbachia load under different stage transcriptome-based constraints.

As *Wolbachia* population dynamics have been well-studied in *B. malayi* (McGarry et al., 2004; Grote et al., 2017), we agree with the reviewer that one source of additional validation might be to calibrate our model to life stage-specific population sizes. However, it should be appreciated that our model is concerned with the total metabolic capacity of the bacterium which does not directly correlate with population size. Further, there is very little information available on the nutrients available in the environments occupied by many of the life stages, rendering it highly challenging to accurately model metabolically inert stages like L3 (Li et al., 2009). We include the above as new text in the manuscript:

“As *Wolbachia* population dynamics have been well-studied in *B. malayi* (McGarry et al., 2004; Grote et al., 2017), it is tempting to attempt to calibrate our model to life stage-specific population sizes. However, our model is concerned with the total metabolic capacity of the bacterium which may not correlate directly with population size, and there is very little known about the nutrients available in the environments occupied by many of the life stages, especially metabolically inert stages like L3 (Li et al., 2009).”

2) A few more cases of FBA's (and pFBA's) demonstrated utility in uncovering real biologically relevant insights should be included in the Introduction along with corresponding references.

We now provide more background and references on the application of FBA in the Introduction:

“Beyond the identification of essential genes and potential therapeutic targets, the analyses of metabolic reconstructions with FBA have been used to identify knowledge gaps and improve annotations in pathogens like *Pseudomonasaeruginosa* (Oberhardt et al., 2008) and *Leishmania major* (Chavali et al., 2008), improve bioreactor yields of non-vital compounds in *Pseudomonasputida* (Puchałka et al., 2008), explain the complex observed substrate specificities of Desulfovibrio vulgaris (Flowers et al., 2018), explain observed metabolic changes in the brains of patients with Parkinson’s disease (Supandi and van Beek, 2018), and even demonstrate the non-biomass related factors affecting tissues growing by cell expansion in tomato plants (Shameer et al., 2020).”